# Unifying Dataset Pruning and Distillation
# for Efficient Large-scale Compression

**Lingao Xiao** [1 2 3]  **Songhua Liu** [3]  **Yang He** [1 2 3]  **Xinchao Wang** [3]

## Abstract

Dataset pruning (DP) and dataset distillation (DD) fundamentally differ in their outputs: DP selects original image subsets, while DD generates synthetic images. Recently, DD's increasing reliance on original images suggests a convergence of the two directions. To investigate this convergence trend, we propose a unified dataset compression (DC) benchmark. This benchmark reveals an interesting trade-off for soft-label-DD: while soft labels provide valuable information, they can make the distillation process less essential, as distilled images may not always outperform random subsets. In addition, the benchmark reveals that in current stages, dataset pruning outperforms dataset distillation at small dataset sizes. Given these observations, we explore hard-label-DC as a complementary approach that emphasizes image quality while offering substantial storage efficiency. Our PCA (Prune, Combine, and Augment) is the first framework that does not rely on soft labels but instead focuses on image quality. (1) "P" means selecting easy samples based on dataset pruning metrics, (2) "C" indicates combining these samples effectively, and (3) "A" is to apply constrained image augmentation during training. Extensive experiments validate that PCA significantly outperforms existing DD and DP methods without soft labels. Code is at GitHub.

## 1. Introduction

Modern dataset compression comes in two main types: *dataset pruning* (DP) (Toneva et al., 2019; Paul et al., 2021; Yang et al., 2023; Zheng et al., 2023), which selects a subset of original images, and *dataset distillation* (DD) (Wang et al., 2018; Cazenavette et al., 2022; Yin et al., 2023; Xiao and He, 2024), which creates synthetic images. While both approaches aim to make datasets smaller, they've been used for different pruning ratios. Dataset distillation creates very small datasets, often keeping just 10-100 images per class (IPC), which is more than 90% smaller than the original. Dataset pruning often only removes less than 50% of images while maintaining good performance.

Interestingly, recent DD methods increasingly rely on original images for better performance, making them more similar to DP methods. Specifically, early works (Yin et al., 2023; Yin and Shen, 2024; Shao et al., 2024a; Xiao and He, 2024) gradually optimize random noise to create synthetic images, and more recently, DWA (Du et al., 2024) initializes synthetic images with real images, and RDED (Sun et al., 2024) uses real image patches to create synthetic images in an optimization-free manner. The recent convergence of DD and DP methods motivates our investigation of their comparative effectiveness.

However, there are two significant limitations that hinder unifying DD and DP into a unified approach. **1) Different reliance on soft label.** DD's reliance on soft labels introduces significant storage, while DP methods avoid such dependencies. Soft labels' storage requirements are excessive – consuming up to 40 times more storage than the images themselves (Xiao and He, 2024). The complexity increases further when incorporating advanced augmentation techniques like those in DELT (Shen et al., 2024). **2) Different evaluation settings.** Different configurations including batch sizes, loss type, and augmentation parameters, largely affect the evaluated performance.

We propose a unified Dataset Compression (DC) benchmark to fairly evaluate DP and DD. The benchmark includes 1) real, 2) partially real, or 3) completely distilled images. We use prefixes to denote whether soft labels are utilized (soft-label-DC) or not (hard-label-DC). Our benchmark reveals a surprising paradox with DD methods: the DD images perform worse than simply keeping random subsets of original images when both are equipped with soft labels as shown in Figure 1, particularly more pronounced at large IPCs. In ad-

[1] CFAR, Agency for Science, Technology and Research, Singapore [2] IHPC, Agency for Science, Technology and Research, Singapore [3] Department of Electrical and Computer Engineering, National University of Singapore, Singapore. Correspondence to: Yang He <he_yang@a-star.edu.sg>.

*Proceedings of the 43rd International Conference on Machine Learning*, Seoul, South Korea. PMLR 306, 2026. Copyright 2026 by the author(s).

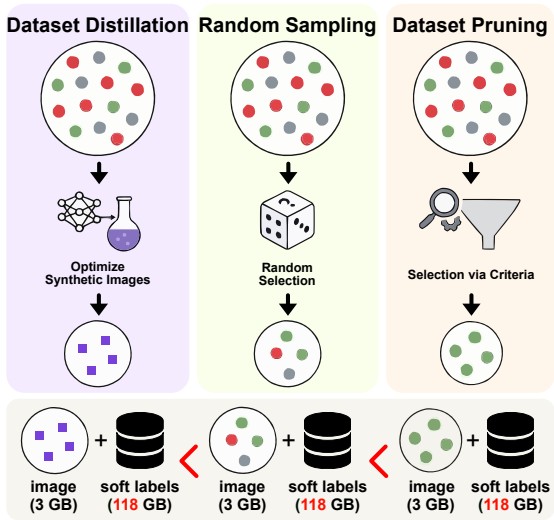

Figure 1. Paradox in Soft-label Dataset Distillation (DD): DD images < random subsets < pruning-based subsets.

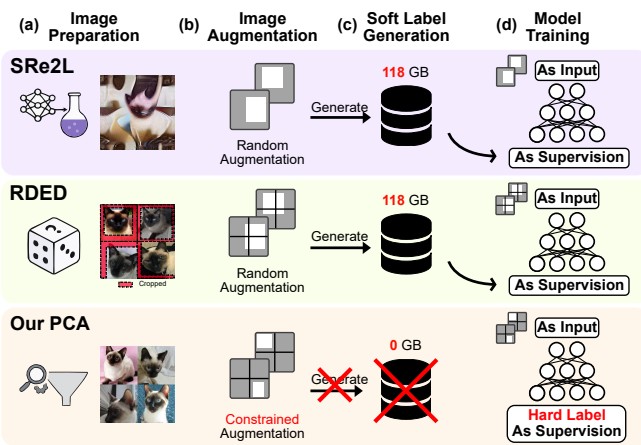

Figure 2. Unlike previous methods (SRe²L (Yin et al., 2023) and RDED (Sun et al., 2024)), our PCA framework introduces four key innovations: (a) pruning-based subset selection, (b) constrained augmentation, (c) elimination of the soft label generation process, and (d) supervision using hard labels instead of soft labels.

dition, even purely random noise achieves learnable results using soft labels from a pretrained teacher network. These discoveries question the validity of current DD methods and raise fundamental concerns about whether focusing on soft labels over images for dataset compression makes sense.

To resolve the paradox in current DD methods, we advocate hard-label-DC and propose the first framework called "Prune, Combine, and Augment (PCA)" that prioritizes image contributions without relying on soft labels, as illustrated in Figure 2. Our PCA framework has four key innovations compared with previous methods. a) **Image Preparation:** PCA leverages pruning insights by selecting simple and representative images based on established pruning principles, then combines them in a cropping-free manner for further compression. b) **Image Augmentation:** PCA applies constrained augmentation to images to adhere to data-scaling laws for the final small-scale datasets during model training. c) **Soft Label Generation:** Unlike prior approaches, PCA does not rely on soft labels generated from pretrained models. d) **Model Training:** PCA uses hard labels exclusively for model supervisions. By avoiding soft labels, PCA is well-suited for scenarios with limited memory, storage, or restricted access to large teacher models. In summary, our primary contributions include:

1. A unified dataset compression benchmark for DD and DP. Three key observations from the benchmark urge us to propose hard-label-DC.

2. The first hard-label-DC framework, PCA (Prune, Combine, Augment), that eliminates dependency on soft labels while focusing on image contributions.

3. Extensive experiments validating PCA's superior performance over both DD and DP methods, showing the power of shifting focus from labels to images.

## 2. Related Works

**Dataset Distillation.** Dataset distillation aims to learn compact and synthetic datasets that achieve similar performance to the full dataset. Researchers have developed many frameworks (Wang et al., 2018; Zhao et al., 2021; Kim et al., 2022; Zhao and Bilen, 2021; Cazenavette et al., 2022; Liu et al., 2023; Lee et al., 2022; Zhao and Bilen, 2023; Wang et al., 2022; Jiang et al., 2022; Du et al., 2023; Shin et al., 2023; Deng and Russakovsky, 2022; Liu et al., 2022a; Zhao and Bilen, 2022; Wang et al., 2023; Lorraine et al., 2020; Nguyen et al., 2021a;b; Vicol et al., 2022; Zhou et al., 2022; Loo et al., 2022; Zhang et al., 2023; Cui et al., 2023; Loo et al., 2023) to effectively learn the synthetic dataset on small-scale datasets like MNIST and CIFAR.

However, scaling the existing framework to a large dataset suffers from unaffordable consumption in both memory and time. SRe²L (Yin et al., 2023) for the first time achieves noticeable performance by decoupling the optimization process into three phases of squeezing, recovering, and relabeling. Follow-up works (Yin and Shen, 2024; Sun et al., 2024; Du et al., 2024; Shao et al., 2024a; Loo et al., 2024) mostly focus on addressing the diversity issue of the recovery phase, with more and more attention paid to the relabeling process (Xiao and He, 2024; Zhang et al., 2024a; Qin et al., 2024a; Kang et al., 2024; Yu et al., 2025). However, most methods use different evaluation settings without direct comparison, and overlook the random baseline's performance under relabeling.

**Dataset Pruning.** Dataset pruning selects a representative subset by ranking images with different metrics (Coleman et al., 2020; Toneva et al., 2019; Pleiss et al., 2020; Feldman and Zhang, 2020; Paul et al., 2021). Most of the reported experiments are focused on small datasets like CIFAR or

*Table 1.* Inconsistent settings and requirements of dataset compression methods. [†] denotes actual image storage is affected by JPEG compression; [*] indicates resizing image to 224x224. IPC-10, ImageNet-1K.

| | Settings/ Requirements | EL2N | RDED | SRe$^2$L |
|---|---|---|---|---|
| Image | DP/DD | DP | DD | DD |
| | Real/Distilled | Real | Partly Real | Distilled |
| | Storage[†] | 118M[*] | 130M | 157M |
| Soft Label | Storage Overhead | - | 5,879M ($\uparrow$**45**$\times$) | 5,822M ($\uparrow$**37**$\times$) |
| | Time Overhead | - | 25 mins ($\uparrow$**1.7**$\times$) | 25 mins ($\uparrow$**1.6**$\times$) |
| Model Training | Batch Size | 256 | 128 | 1024 |
| | Num. of Iterations | 300K | 24K | 24K |

ImageNet subsets. Methods that scale to large-scale datasets focus on small or moderate pruning ratios to ensure minimum performance drop (Xia et al., 2023; Sorscher et al., 2022; Zheng et al., 2023; Zhang et al., 2024b; Grosz et al., 2024; Abbas et al., 2024). VID (Ben-Baruch et al., 2024) conducts experiments on data pruning methods using knowledge distillation. However, these experiments did not explore extreme pruning ratios, and the baselines were not compared with dataset distillation methods.

**Combining Dataset Distillation and Dataset Pruning.** Dataset compression intuitively encompasses both dataset distillation and dataset pruning, which can work independently. Existing studies incorporate the pruning or coreset selection before dataset distillation (Liu et al., 2023; Xu et al., 2025; Moser et al., 2024; Shen et al., 2024). YOCO (He et al., 2024) examines the pruning rules specifically for distilled datasets. Concurrent to our work, Li et al. rethink the evaluation of dataset distillation. However, given the distinctly different nature and settings of these two tasks, it remains unclear which method represents the state-of-the-art (SOTA) in the field of data compression today. This lack of direct comparison may lead to misunderstandings about the data compression task (which is also discussed by Dey et al.) and result in ineffective combinations of methods.

## 3. Benchmarking Data Compression

**DD and DP's Difference 1: Soft Label.** As shown in Table 1, DD methods typically use soft labels, while DP methods exclusively use hard labels. However, as mentioned by Xiao and He and Qin et al., the soft label storage far exceeds the image storage. For example, the label storage of ImageNet-10 IPC10 is over 5.8 GB, while the images are merely 157M, creating a 40$\times$ storage gap. Existing methods (Xiao and He, 2024; Zhang et al., 2024a) have started to reduce soft label storage, but pre-generated soft labels still face several disadvantages. (1) Soft labels are stored in a very different format from images, and special changes to the dataloader are required; (2) despite being GPU-compute intensive, the soft label generation process has significant memory-transfer bottlenecks, being unfriendly to devices with limited CPU resources. Last but not least, as more and more data augmentation is introduced, (3) the use of soft labels becomes increasingly complicated as more advanced augmentation (i.e., RandAugment (Shen et al., 2024)) is introduced; (4) soft label introduces knowledge beyond the compressed datasets, potentially biasing the evaluation.

**DD and DP's Difference 2: Inconsistent Hyperparameters.** First, DD and DP methods have different hyper-parameters as shown in Table 1. DP methods often train for a fixed number of iterations as in full dataset training, while DD methods train a fixed number of epochs regardless of the dataset size. Second, DD methods themselves have varying evaluation settings (see Appendix B.2), where SRe$^2$L (Yin et al., 2023) initially used a batch size of 1024 while later studies (Yin and Shen, 2024; Du et al., 2024; Sun et al., 2024) employed much smaller batches, dramatically affecting performance. These differences create barriers to reproducibility and complicate meaningful cross-method comparisons. We adopt CDA's setting (Yin and Shen, 2024) as our standard evaluation protocol for both DD and DP methods since it's widely used.

**Our Dataset Compression (DC) Benchmark to unify DD and DP.** For fair comparison and training efficiency, we standardize all experiments using the most common evaluation protocol from dataset distillation (CDA (Yin and Shen, 2024)), for both DD and DP methods. To ensure comparability, we keep the training setup identical across all experiments, varying only the input dataset. Additionally, to match the fixed image resolution required by DD methods, we preprocess DP images by cropping along the shorter side and resizing them to 224$\times$224.

**Benchmark Observation 1: (DD + Soft Label) < (Random + Soft Label).** Existing dataset distillation (DD) methods do not consider random subsets as a baseline. However, after benchmarking random subsets under the standard evaluation setting with soft labels, we found that most dataset distillation methods (Yin et al., 2023; Yin and Shen, 2024; Du et al., 2024; Xiao and He, 2024) fail to surpass the random baseline, especially at large IPCs as shown in Figure 3a. The high random baseline with soft labels reveals that **the inflated performance gain of DD methods is primarily due to the soft labels**.

**Benchmark Observation 2: (Random + Soft Label) < (Pruning + Soft Label).** An important research question remains unanswered: *how do DP methods perform with soft labels at the extreme pruning ratios typical of DD methods?* Our benchmark demonstrates that DP methods consistently outperform random subsets when soft labels are applied to DP. This indicates that pruned datasets are more effective than random subsets and DD datasets. This observation

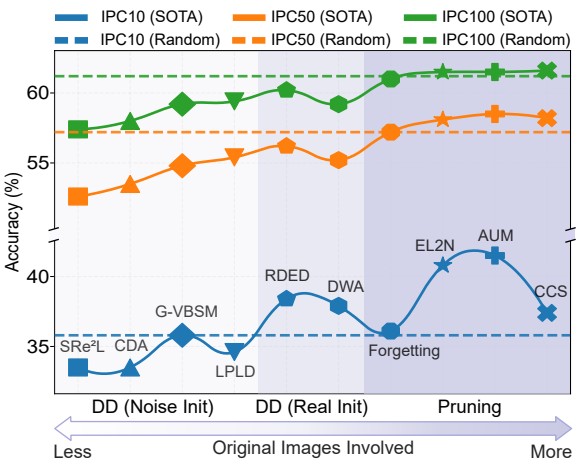 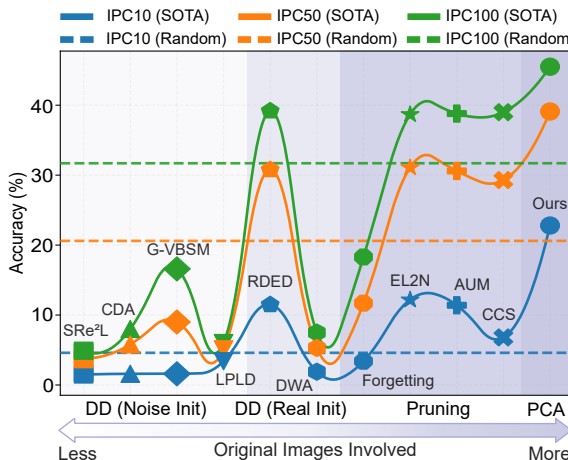

*(a)* Benchmarking SOTA methods using **soft labels**. Detailed data is provided in Table 2a.

*(b)* Benchmarking SOTA methods using **hard labels**. Detailed data is provided in Table 2b.

*Figure 3.* Comparison of SOTA methods using soft labels (left) and hard labels (right) on ImageNet-1K. "DD (Noise Init)" and "DD (Real Init)" denote dataset distillation initialized with noisy images and real images, respectively. Evaluation uses ResNet-18 on ImageNet-1K. Two observations are made: (1) Many methods struggle to outperform the random baseline, particularly at large IPCs. (2) In addition, methods utilizing more original images generally achieve better performance.

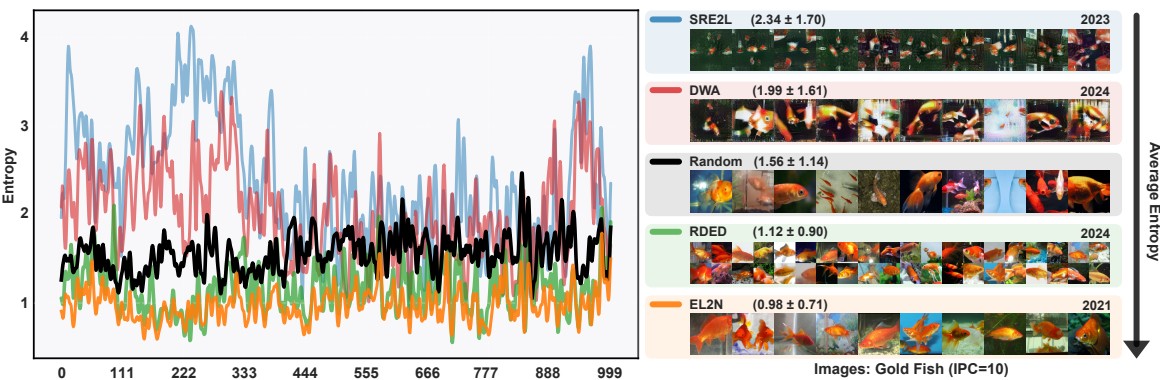

*Figure 4.* Entropy analysis of different datasets with IPC=10. Images are randomly sampled from the corresponding dataset for visualization. The classifier used for entropy analysis is the pretrained EfficientNet-B0 (Tan and Le, 2019).

also helps explain why recent DD methods increasingly incorporate high-quality original images.

**Benchmark Observation 3: DD < Random < Pruning.** Given the substantial storage and computational overhead of soft labels, we investigated whether the performance trends would hold when using only hard labels. Our experiments show this trend persists with hard labels (Figure 3b), which are more practical due to lower storage requirements. With hard labels only, the performance gap between methods widens, confirming that pruning's advantages stem from image quality, not soft label utilization. This further validates that **previously observed DD advantages were primarily due to soft labels, not the distilled images themselves**.

These three observations, combined with the substantial storage overhead of soft labels, suggest that large-scale dataset compression should prioritize image quality over

soft label exploitation. To this end, we are motivated to develop a hard-label-only framework that shifts focus from labels to images.

## 4. Framework: Prune, Combine, and Augment

Figure 5 shows our *Prune, Combine, and Augment (PCA)* framework, which removes soft labels and supervises models with hard labels.

### 4.1. Prune Dataset

**Motivation.** Section 3 demonstrates that pruning consistently outperforms distillation. Based on this finding, we incorporate dataset pruning into our framework by leveraging three key insights: (1) Class balance becomes increasingly critical as dataset size diminishes (He et al., 2024),

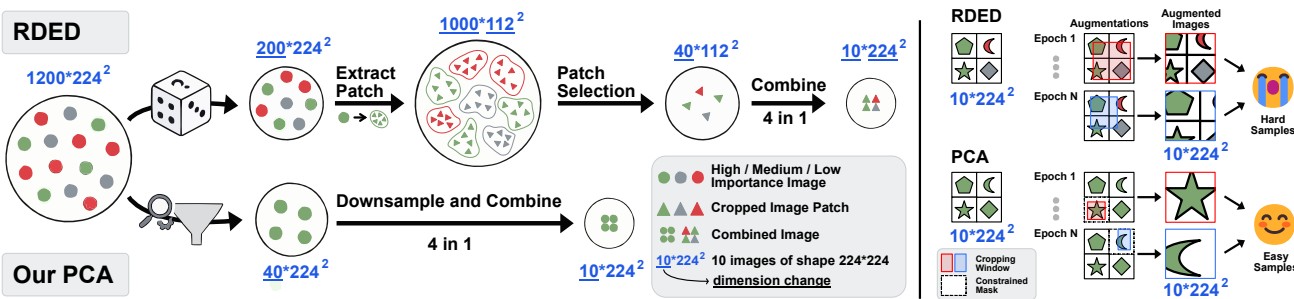

*Figure 5.* Detailed PCA pipeline. **Left**: illustration of image preparation, where our PCA includes only high-importance images. **Right**: illustration of image augmentation, where our PCA constrains image cropping at a single image patch, creating easy samples favored by the data-scaling law.

(2) Simpler images yield better performance with small datasets (Sorscher et al., 2022; Zheng et al., 2023; He et al., 2024), and (3) Pruning should be applied to the complete dataset.

**Insight 1: Maintain Perfect Class Balance in Pruning.** Conventional dataset pruning creates an imbalanced dataset where less important classes are pruned more aggressively. At extreme pruning ratios, this can completely eliminate certain classes (He et al., 2024). In contrast, dataset distillation maintains perfect class balance by generating a fixed number of images per class (IPC). We adopt this balanced approach by pruning to a consistent IPC across all classes.

**Insight 2: Prioritize Simpler Images During Pruning.** Prior research (Zheng et al., 2023; He et al., 2024) demonstrates that simpler images perform better when the dataset size is small. Our entropy analysis in Figure 4 provides an intuitive explanation for why pruning methods outperform distillation methods. By measuring dataset complexity through entropy (Coleman et al., 2020; Sun et al., 2024), we observe that pruned datasets have the lowest average entropy, indicating relative simplicity. Visual inspection confirms that images retained by pruning methods are indeed simpler than those created by distillation methods. Based on these findings, we follow He et al. in using the reversed EL2N metric (Paul et al., 2021) for our pruning strategy.

**Insight 3: Apply Pruning to the Full Dataset.** Without soft labels, maximizing information retention becomes crucial. Therefore, pruning must be conducted on the full dataset rather than on subsets. As shown in Figure 5 (right), our approach differs from methods like RDED (Sun et al., 2024), which creates image patches from randomly sampled subsets. Instead, we prune the complete dataset to ensure all subsequent operations work exclusively with the most informative samples.

### 4.2. Cropping-Free Image Combination

In our PCA framework, where only hard labels are available and the dataset has already been carefully pruned, cropping or patch-based selection is unsuitable. This is because pruning makes the dataset retain only the most important images; any further cropping risks irreversibly discarding important content that hard-label supervision cannot recover.

To formalize this, we clarify the relationships between negative log-likelihood (NLL), cross-entropy, and entropy. For a dataset $\mathcal{D} = \{(x_i, y_i)\}_{i=1}^N$ and model $p_\theta(y|x)$:

$$\text{NLL}(\mathcal{D}; \theta) = -\frac{1}{N} \sum_{i=1}^N \log p_\theta(y_i|x_i) \approx \text{CE}_{\mathcal{D}}(p_{\text{true}}, p_\theta)$$

$$= H_{\text{true}} + D_{\text{KL}}(p_{\text{true}} \| p_\theta),$$

where $H_{\text{true}}$ is the irreducible entropy of the true conditional distribution. For our analysis, we focus on the model's predictive entropy:

$$H(\mathcal{D}; \theta) = \frac{1}{N} \sum_{i=1}^N H(p_\theta(\cdot|x_i)),$$

$$\text{where} \quad H(p_\theta(\cdot|x)) = -\sum_{y=1}^C p_\theta(y|x) \log p_\theta(y|x).$$

While ideally one would train separate models on each dataset subset, we use a fixed pretrained model $\theta_0$ as an efficient proxy for evaluation, as pretrained model uncertainty correlates strongly with dataset difficulty and trainability (Coleman et al., 2020). This allows us to write $\text{NLL}(\mathcal{D}) := \text{NLL}(\mathcal{D}; \theta_0)$ and $H(\mathcal{D}) := H(\mathcal{D}; \theta_0)$ for brevity.

Let $\mathcal{C}_{\text{sel}}(\mathcal{D})$ denote selective cropping that chooses optimal crops for each image to minimize NLL, and let $\mathcal{A}_r(\mathcal{D})$ denote random spatial cropping augmentation with ratio $r \in (0, 1]$, where $r$ represents the fraction of area retained. We reveal two fundamental limitations of cropping-based approaches:

**Proposition 4.1** (proof in Appendix A.1). *Let $\mathcal{D}' = \mathcal{C}_{\text{sel}}(\mathcal{D})$ be a selectively cropped version of dataset $\mathcal{D}$. Lower evaluation loss does not guarantee lower entropy:*

$$\text{NLL}(\mathcal{D}') < \text{NLL}(\mathcal{D}) \;\not\Rightarrow\; H(\mathcal{D}') < H(\mathcal{D}).$$

*Table 2.* Benchmarking SOTA methods against random baseline under evaluation with **soft labels** (top) and **hard labels** (bottom). † means optimization-free distillation approaches. All experiments use ResNet-18 on ImageNet-1K. Tables with standard deviation are provided in Appendix C.

*(a)* Soft label benchmark. (**Storage overhead of soft labels:** $\sim 40\times$ **as images.**)

| IPC | Random | DD (Noise Init) | | | | DD (Real Init) | | Pruning Method with Rules | | | |
|---|---|---|---|---|---|---|---|---|---|---|---|
| | | SRe²L | CDA | G-VBSM | LPLD | RDED† | DWA | Forgetting | EL2N | AUM | CCS |
| 10 | $35.8_{\pm0.2}$ | $33.5_{\downarrow2.3}$ | $33.5_{\downarrow2.3}$ | $35.8_{=0.0}$ | $34.6_{\downarrow1.2}$ | $38.4_{\uparrow2.6}$ | $37.9_{\uparrow2.1}$ | $36.1_{\uparrow0.3}$ | $40.8_{\uparrow5.0}$ | $\mathbf{41.5}_{\uparrow5.7}$ | $37.4_{\uparrow1.6}$ |
| 50 | $57.2_{\pm0.2}$ | $52.6_{\downarrow4.6}$ | $53.5_{\downarrow3.7}$ | $54.8_{\downarrow2.4}$ | $55.4_{\downarrow1.8}$ | $56.2_{\downarrow1.0}$ | $55.2_{\downarrow2.0}$ | $57.2_{=0.0}$ | $58.1_{\uparrow0.9}$ | $\mathbf{58.5}_{\uparrow1.3}$ | $58.2_{\uparrow1.0}$ |
| 100 | $61.2_{\pm0.2}$ | $57.4_{\downarrow3.8}$ | $58.0_{\downarrow3.2}$ | $59.2_{\downarrow2.0}$ | $59.4_{\downarrow1.8}$ | $60.2_{\downarrow1.0}$ | $59.2_{\downarrow2.0}$ | $61.0_{\downarrow0.2}$ | $61.5_{\uparrow0.3}$ | $61.5_{\uparrow0.3}$ | $\mathbf{61.6}_{\uparrow0.4}$ |

*(b)* Hard label benchmark and Our PCA. (**No storage overhead of soft labels for all IPC.**)

| IPC | Random | DD (Noise Init) | | | | DD (Real Init) | | Pruning Method with Rules | | | | PCA |
|---|---|---|---|---|---|---|---|---|---|---|---|---|
| | | SRe²L | CDA | G-VBSM | LPLD | RDED† | DWA | Forgetting | EL2N | AUM | CCS | Ours |
| 10 | $4.6_{\pm0.1}$ | $1.5_{\downarrow3.1}$ | $1.6_{\downarrow3.0}$ | $1.6_{\downarrow3.0}$ | $3.4_{\downarrow1.2}$ | $11.5_{\uparrow6.9}$ | $1.9_{\downarrow2.7}$ | $3.4_{\downarrow1.2}$ | $12.2_{\uparrow7.6}$ | $11.4_{\uparrow6.8}$ | $6.8_{\uparrow2.2}$ | $\mathbf{22.8}_{\uparrow18.2}$ |
| 50 | $20.6_{\pm0.1}$ | $3.8_{\downarrow16.8}$ | $5.8_{\downarrow14.8}$ | $9.0_{\downarrow11.6}$ | $5.1_{\downarrow15.5}$ | $30.8_{\uparrow10.2}$ | $5.3_{\downarrow15.3}$ | $11.7_{\downarrow8.9}$ | $31.1_{\uparrow10.5}$ | $30.6_{\uparrow10.0}$ | $29.3_{\uparrow8.7}$ | $\mathbf{39.1}_{\uparrow18.5}$ |
| 100 | $31.7_{\pm0.6}$ | $4.9_{\downarrow26.8}$ | $8.0_{\downarrow23.7}$ | $16.6_{\downarrow15.1}$ | $6.0_{\downarrow25.7}$ | $39.2_{\uparrow7.5}$ | $7.5_{\downarrow24.2}$ | $18.3_{\downarrow13.4}$ | $38.7_{\uparrow7.0}$ | $38.8_{\uparrow7.1}$ | $39.0_{\uparrow7.3}$ | $\mathbf{45.5}_{\uparrow13.8}$ |

**Theorem 4.2** (proof in Appendix A.2)**.** *Let* $\mathcal{D}' = \mathcal{C}_{sel}(\mathcal{D})$ *be a selectively cropped dataset with lower initial entropy:* $H(\mathcal{D}') < H(\mathcal{D})$. *There exists a crop ratio* $r^* \in (0,1)$ *such that when random cropping augmentation is applied, the entropy advantage is lost:*

$$H(\mathcal{D}') < H(\mathcal{D}) \quad but \quad H(\mathcal{A}_{r^*}(\mathcal{D}')) \geq H(\mathcal{A}_{r^*}(\mathcal{D})),$$

*where* $H(\mathcal{A}_r(\cdot))$ *represents the expected entropy over all random spatial crops with ratio* $r$.

**Interpretation.** These results demonstrate a two-fold limitation: (1) cropping to lower NLL doesn't necessarily reduce dataset entropy, which is what matters for performance; and (2) even if entropy is reduced through selective cropping, this advantage is lost or reversed when training-time augmentations are applied. This theoretical analysis, combined with our empirical findings, justifies our choice to avoid cropping and instead combine full, pruned images, ensuring maximal information retention and reliable downstream performance.

### 4.3. Constrained Augmentation for Data-Scaling-Law

The scaling-law usually refers to scaling up the model (Kaplan et al., 2020); however, we refer to the data-scaling-law (Sorscher et al., 2022) which scales the dataset, specifically when scaling down under hard-label-only settings. After acquiring a small-scale dataset, it remains crucial to unveil its potential and effectively harness the available information. Augmentation typically serves as the tool to achieve the goal, but it is imperative that augmentation outcomes should closely adhere to the data-scaling-law. For example, RDED (Sun et al., 2024) selects simple image patches and combines them; however, during training, the `Random Resized Crop` operation directly applies to the combined image, inadvertently transforming simple images into more complex ones.

To counteract this issue, we propose to randomly restrict the cropping area within a single patch, and we refer to it

as constrained augmentation. The illustration is provided in Figure 5 (right). Our constrained augmentation uses a single augmented image instead of four per epoch for training. Therefore, no additional training overhead is imposed when compared to RDED (Sun et al., 2024).

We emphasize the importance of using an effective augmentation strategy. When dealing with a small number of images, achieving good performance can be challenging. A well-crafted augmentation method, which adheres to data-scaling-law, can greatly enhance the potential of the images.

## 5. Experiment

All experiments are conducted on ImageNet-1K using CDA's evaluation settings (see Appendix B.2) unless otherwise indicated. Additional settings, including dataset, networks, and baseline specifications, can be found in Appendix B.

### 5.1. Primary Results

**Call Attention to Pruning from Soft-label Benchmark.** Table 2a benchmarks existing dataset distillation methods and dataset pruning methods under the same evaluation setting. We notice that by increasing the batch size in the evaluation setting, the performance of SRe²L (Yin et al., 2023) catches up with other SOTA methods (Yin and Shen, 2024; Xiao and He, 2024). However, with this being said, many SOTA methods cannot beat the random baseline. Surprisingly, pruning methods that are published 3-5 years ago (Toneva et al., 2019; Pleiss et al., 2020; Paul et al., 2021) unanimously outperform random baselines, and it's time to call attention to this under-explored topic. As a result, an interesting observation is that the performance improves as the images include more prior knowledge of original datasets.

**Comparing Hard-label SOTA Methods with PCA.** Table 2b evaluates the SOTA methods using a more advocated

*Table 3.* Performance of pruning methods at extreme pruning ratio. The best setting for each method is marked in **bold**, and the best method is underlined.

| Method | Soft Label | | | | | | | | Hard Label | | | | | | | |
|---|---|---|---|---|---|---|---|---|---|---|---|---|---|---|---|---|
| | IPC10 (99.22%) | | | | IPC50 (96.97%) | | | | IPC10 (99.22%) | | | | IPC50 (96.97%) | | | |
| | hard | hard$_B$ | easy | easy$_B$ | hard | hard$_B$ | easy | easy$_B$ | hard | hard$_B$ | easy | easy$_B$ | hard | hard$_B$ | easy | easy$_B$ |
| Forgetting | 25.9 | 32.9 | 6.1 | **36.1** | 53.0 | 56.7 | 52.3 | **57.2** | 0.4 | **4.4** | 0.1 | 3.4 | 15.3 | **21.7** | 0.3 | 11.6 |
| AUM | 27.1 | 37.4 | 12.2 | **41.5** | 53.7 | 56.8 | 45.3 | **58.5** | 0.2 | 1.4 | 0.1 | **11.4** | 1.8 | 4.4 | 0.3 | **30.6** |
| EL2N | 28.7 | 36.0 | 14.2 | **40.8** | 54.4 | 56.9 | 46.0 | **58.1** | 0.2 | 1.4 | 0.2 | **12.2** | 3.2 | 4.2 | 0.3 | **31.1** |

*Table 4.* Hard label performance against soft labels. * denotes hard-label only. ResNet-18, ImageNet-1K.

| Compression Rate | 30× SRe²L | 40× CDA | 100× LPLD | > 300× PCA* |
|---|---|---|---|---|
| IPC10 | 14.1 | 13.2 | 9.6 | 25.6 |
| IPC50 | 37.2 | 38.0 | 33.7 | 42.4 |
| IPC100 | 46.7 | 47.2 | 44.7 | 48.8 |

*Table 5.* Ablation study of the proposed PCA framework. + denotes add-on components. Note that the default augmentation applies unless marked with †, denoting the proposed constrained augmentation. Best results of each setting are in **bold**. ResNet-18, ImageNet-1K.

| Setting | Method | 10 | 50 | 100 |
|---|---|---|---|---|
| AdamW | Random | 4.6 | 21.2 | 31.4 |
| | + Pruning | 12.2↑7.6 | 31.1↑9.9 | 38.8↑7.4 |
| | + Combine | 14.4↑9.8 | 32.4↑11.2 | 39.4↑8.0 |
| | + Augment† | 22.8↑18.2 | 39.1↑17.9 | 45.5↑14.1 |
| SGD | Random | 5.1 | 26.6 | 38.9 |
| | Our PCA | **25.6**↑20.5 | **42.1**↑15.5 | **48.6**↑9.7 |

*Table 6.* Cross-architecture performance of PCA framework (hard-label) on ImageNet-1K. "→ SGD" denotes SGD setting.

| Model | Params. | Acc. | 10 | 50 | 100 |
|---|---|---|---|---|---|
| ResNet-18 → SGD | 11.7 M | 69.76 | 22.8 / 25.6 | 39.1 / 42.1 | 45.5 / 48.6 |
| ResNet-50 → SGD | 25.6 M | 76.13 | 23.0 / 25.3 | 42.3 / 43.2 | 48.3 / 50.5 |
| ResNet-101 → SGD | 44.5 M | 77.37 | 25.8 / 25.9 | 42.7 / 46.3 | 49.6 / 53.6 |
| MobileNet-V2 | 3.5 M | 71.88 | 21.9 | 39.1 | 45.3 |
| EfficientNet-B0 | 5.3 M | 77.69 | 25.0 | 42.4 | 50.4 |
| Swin-V2-Tiny | 28.4 M | 82.07 | 15.3 | 37.8 | 48.2 |

*Table 7.* Effects of regularization-based augmentations on PCA (SGD setting). "Crop" = *RandomResizedCrop*. "Label Mixing" = whether to mix class labels. ResNet-18, IPC10, ImageNet-1K.

| Crop | Data Mixing | Label Mixing | Mix Probability | | |
|---|---|---|---|---|---|
| | | | 0.2 | 0.5 | 1.0 |
| ✓ | ✗ | - | | 25.6 | |
| ✓ | CutMix | ✓ | 23.8↓1.8 | 23.0↓2.6 | 17.4↓8.2 |
| ✓ | CutMix | ✗ | 25.5↓0.1 | 24.7↓0.9 | 23.0↓2.6 |
| ✓ | Mixup | ✓ | 25.7↑0.1 | 23.0↓2.6 | 7.7↓17.9 |
| ✓ | Mixup | ✗ | 25.9↑0.3 | 25.1↓0.5 | 17.6↓8.0 |
| ✓ | **Cutout** | - | **26.2**↑0.6 | **25.7**↑0.1 | 25.3↓0.3 |

approach that does not introduce any additional storage costs besides the images or requires pretrained knowledge. By utilizing only hard labels, most results show a similar trend to soft label benchmarks. Our PCA (**P**rune, **C**ombine, and **A**ugment) framework essentially exceeds the random baseline and other SOTA methods at all tested IPCs.

**Sanity Check on Pruning Rules and Scaling Laws.** Previous pruning methods (Sorscher et al., 2022; Zheng et al., 2023; He et al., 2024) concluded that with small datasets, (1) easy images are preferred and (2) class balance is important. However, these findings need verification in our extreme pruning scenario (IPC10 = 99.2% pruning rate) since prior works (Zheng et al., 2023) used more moderate ratios or focused on distilled datasets (He et al., 2024). Table 3 confirms these rules hold even at extreme pruning ratios with real images, as selecting easy images with balanced classes consistently delivers the best results under both soft and hard label settings. Among pruning metrics, EL2N (Paul et al., 2021) shows superior performance and requires less computation time, making it our chosen method for PCA (see Appendix D.2 for analysis of why Forgetting (Toneva et al., 2019) performs worse).

## 5.2. More Experiments

**Ablation Study.** Table 5 demonstrates the improvements contributed by each component under hard-label-only settings. Every component in PCA is essential and advantageous to the final performance. Especially, constrained augmentation has the most impact on the final performance. This validates our design principle that adhering to the data-scaling-law is crucial. Additionally, since the evaluation

settings for pruning methods use SGD with an initial learning rate of 0.1, we have also conducted evaluations under this configuration. Our observations reveal that even random baselines benefit from using SGD (0.1), showing a distinct advantage over AdamW (0.01). In all cases, the proposed PCA framework yields significant improvements over random baselines.

**Performance Against Soft Labels.** Despite having inevitable drawbacks and unfairness as mentioned in Section 3, the cumbersome storage of soft labels can be addressed to some degree. Table 4 shows our hard-label-only framework can perform on par or even surpass previous methods using part of soft labels. In theory, the maximum soft label compression rate is limited to 300× in ImageNet-1K setting, as each image requires a soft label per epoch for 300 epochs. Since we do not use soft labels at all, our soft label compression rate is > 300×.

**Cross Architecture Performance.** Table 6 demonstrates a good generalization ability of the proposed framework. For all validation models, the performance scales well with the dataset size. In addition, the framework scales well with improved model capacity, with one exception on the transformer-based Swin-V2-Tiny model (Liu et al., 2022b). Since the transformer-based model is extremely data-hungry, a trend is also observed in previous works (Xiao and He, 2024; Sun et al., 2024).

**Regularization-based Data Augmentation.** In addition to common augmentation techniques such as random resized crop and horizontal flips, data mixing augmentation (i.e., Mixup, Cutout, and CutMix) is a regularization-based data augmentation that reduces overfitting by providing di-

*Table 8.* Dataset cropping configs. $N$ is number of extracted patches.

| Observer | Metric | $N = 5$ | $N = 20$ |
|---|---|---|---|
| EfficientNet-B0 | NLL | 19.0 | 18.3 |
| | Entropy | 17.2 | 18.1 |
| ResNet | NLL | 18.7 | 17.1 |
| | Entropy | 18.0 | 18.3 |
| No Crop | | 22.8 | |

*Table 9.* Crop ratio ($r$) during training.

| range:=$(r, 1.0)$ | IPC10 | IPC50 |
|---|---|---|
| $r = 0.01$ | 22.1 | 39.0 |
| $r = 0.08$ | **22.8** | **39.1** |
| $r = 0.5$ | 22.2 | 38.6 |
| $r = 0.8$ | 21.0 | 35.5 |
| $r = 1.0$ | 18.7 | 34.0 |

*Table 11.* Perfromance on Object Detection. VOC (2007+2012) dataset. * denotes simple adoptation to object detection task, applied with two rules from He et al.. [†] denotes results reported in VPS (Yagi, 2025).

| Method | #. Images | mAP | $AP_{75}$ | $AP_{50}$ |
|---|---|---|---|---|
| Random | 1,000 | $21.17_{\pm0.61}$ | $17.26_{\pm0.85}$ | $44.97_{\pm0.78}$ |
| AUM* | 1,000 | $22.31_{\pm0.04}$ | $18.23_{\pm0.53}$ | $46.36_{\pm0.39}$ |
| VPS[†] | 1,655 | $30.05_{\pm \text{NaN}}$ | $25.84_{\pm \text{NaN}}$ | $60.77_{\pm \text{NaN}}$ |
| PCA (Ours) | **1,000** | $\mathbf{35.99}_{\pm0.33}$ | $\mathbf{34.86}_{\pm0.59}$ | $\mathbf{65.89}_{\pm0.44}$ |
| Full Dataset | 16,551 | 52.16 | 56.92 | 80.63 |

*Table 10.* PCA framework with different pruning methods. ResNet-18 with AdamW optimizer.

| IPC | Random | Forgetting | EL2N | AUM | TDDS |
|---|---|---|---|---|---|
| 10 | $4.6_{\pm0.1}$ | $8.6_{\pm0.2}$ | $\mathbf{22.8}_{\pm0.3}$ | $21.9_{\pm0.3}$ | $20.0_{\pm0.4}$ |
| 50 | $20.6_{\pm0.1}$ | $24.1_{\pm0.4}$ | $39.1_{\pm0.2}$ | $39.2_{\pm0.1}$ | $\mathbf{40.8}_{\pm0.1}$ |
| 100 | $31.7_{\pm0.6}$ | $36.2_{\pm0.3}$ | $45.5_{\pm0.4}$ | $46.4_{\pm0.2}$ | $\mathbf{47.6}_{\pm0.0}$ |

verse and challenging examples during training. Among all options, Table 15 shows that Cutout demonstrates the best performance, while CutMix and Mixup exhibit notable performance degradation as mixing probability increases, especially in the presence of label mixing. This performance advantage is attributed to being best aligned with the scaling law. Details are provided in Appendix D.1.

**Effect of Cropping.** In addition to the theoretical analysis of the effects of cropping (Section 4.2), we conducted experiments to validate our findings. It is important to note that cropping can be performed both before and during training. We refer to cropping the dataset before training a model as dataset cropping, which is irreversible. Table 8 shows that regardless of the metric and observer used to select patches from a well-pruned dataset, dataset cropping negatively impacts performance. This behavior can be explained by Theorem 4.2. Another cropping operation occurs during training augmentation (specifically, RandomResizedCrop), which is "recoverable" because the original image remains unchanged, and the cropping operation in each epoch is independent. Table 9 presents performance under different training crop ratios.

**PCA with Different Pruning Methods.** Table 10 shows PCA results with various pruning methods under hard-label settings. All significantly outperform random baselines. While TDDS (Zhang et al., 2024b) and AUM (Pleiss et al., 2020) show better results at higher IPCs, we choose EL2N (Paul et al., 2021) as our baseline for efficiency, as it requires only 10 epochs of training dynamics compared to 90 epochs required by TDDS and AUM. Forgetting (Toneva et al., 2019), though performing worse than other methods, still consistently beats random baselines. See Appendix D.2 for analysis of Forgetting's limitations.

**Performance on Object Detection Task.** To evaluate the generalizability of our PCA framework beyond image classification, we conduct experiments on the object detection task. Table 11 demonstrates that our method consistently

outperforms all baselines, including VPS (Yagi, 2025), the current state-of-the-art dataset pruning method for object detection. Notably, our method achieves substantial performance gains over VPS while using fewer images. For fair comparison, we follow the experiment settings in VPS (Yagi, 2025) and use Faster R-CNN-C4 (Ren et al., 2015) with a ResNet-50 (He et al., 2016) backbone. PASCAL VOC 2007 and 2012 (Everingham et al., 2010) datasets are used for training, and VOC 2007 test set is used for evaluation. Default training configurations and implementations are based on Detectron2 (Wu et al., 2019). We use AUM (Pleiss et al., 2020) as the pruning method for our PCA framework.

**Additional Discussion (Appendix D).** Additional discussions are provided in Appendix D, including training with purely noisy data (Appendix D.3), SRe$^2$L with real images as initialization (Appendix D.4), random baseline in soft-label-DD (Appendix D.5), relationship between data balance and data stratification (Appendix D.6), mosaic augmentation (Appendix D.7), computation cost and wall-clock overhead analysis (Appendix D.8), and comparison with RDED (Appendix D.9).

**Visualization.** Visualizations of our PCA including baseline methods are provided in Appendix F.

## 6. Conclusion

Our unified dataset compression benchmark revealed a paradox: distilled images with soft labels underperform random subsets, while pruned datasets consistently outperform both, suggesting contemporary DD gains stem from soft labels, which impose up to 40× storage overhead, rather than from distilled images. We address this with our *Prune, Combine, and Augment (PCA)* framework, which selects easy and balanced samples via pruning metrics, combines them effectively, and applies constrained augmentation aligned with data-scaling laws. By using only hard labels, PCA eliminates pretrained resource dependencies and significantly reduces storage requirements while consistently outperforming existing random baselines, particularly at extreme compression ratios.

Limitations and future works are discussed in Appendix G and Appendix H, respectively.

## Acknowledgment

This research is supported by A*STAR Career Development Fund (CDF) under Grant C243512011, the National Research Foundation, Singapore under its National Large Language Models Funding Initiative (AISG Award No: AISG-NMLP-2024-003), and the Ministry of Education, Singapore, under the Academic Research Fund Tier 1 (FY2026, WBS: A-8004345-00-00). Any opinions, findings and conclusions or recommendations expressed in this material are those of the author(s) and do not reflect the views of National Research Foundation, Singapore.

## Impact Statement

This paper addresses pressing challenges in dataset compression by establishing a benchmark for fair comparison across dataset distillation and pruning techniques. By highlighting inconsistencies in previous evaluations, we draw attention to the need for standardized practices that enhance reproducibility and fairness. Our proposed *Prune, Combine, and Augment (PCA)* framework prioritizes image data and utilizes only hard labels, thereby reducing storage and computational demands traditionally associated with soft labels. This approach not only makes dataset compression more practical and accessible but also shifts the research focus back to the images themselves, potentially leading to more balanced and efficient methods. Through these efforts, we aim to foster responsible advancements in large-scale machine learning while ensuring the benefits are accessible to a wider range of practitioners.

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

# Appendix

## A. Proofs

### A.1. Proof of Proposition 4.1

**Proposition A.1** (Restated). *Let $\mathcal{D}' = \mathcal{C}_{sel}(\mathcal{D})$ be a selectively cropped version of dataset $\mathcal{D}$. Lower evaluation loss does not guarantee lower entropy:*

$$\text{NLL}(\mathcal{D}') < \text{NLL}(\mathcal{D}) \;\not\Rightarrow\; H(\mathcal{D}') < H(\mathcal{D}).$$

*Proof.* We prove this by constructing an explicit counterexample using a two-sample dataset and a fixed model $p_{\theta_0}$.

**Counterexample Construction.** Consider a binary classification task with dataset $\mathcal{D} = \{(x_1, y_1), (x_2, y_2)\}$ where $y_1 = y_2 = 1$. Let the model's predictions on the original images be:

$$p_{\theta_0}(y = 1|x_1) = 0.7, \quad H(p_{\theta_0}(\cdot|x_1)) = h(0.7) = 0.6109 \tag{1}$$

$$p_{\theta_0}(y = 1|x_2) = 0.2, \quad H(p_{\theta_0}(\cdot|x_2)) = h(0.2) = 0.5004 \tag{2}$$

where $h(p) = -p \ln p - (1 - p) \ln(1 - p)$ is the binary entropy function using natural logarithm.

The original dataset metrics are:

$$\text{NLL}(\mathcal{D}) = \frac{1}{2}[-\ln(0.7) - \ln(0.2)] = \frac{1}{2}[0.3567 + 1.6094] = 0.9831$$

$$H(\mathcal{D}) = \frac{1}{2}[0.6109 + 0.5004] = 0.5557$$

**Selective Cropping Operation.** Define the selective cropping $\mathcal{C}_{\text{sel}}$ as follows: for $x_1$, retain the original image (no cropping needed as the model already performs well); for $x_2$, apply a crop that removes confounding background elements. Suppose this crop transforms $x_2$ into $x_2'$ such that:

$$p_{\theta_0}(y = 1|x_2') = 0.5, \quad H(p_{\theta_0}(\cdot|x_2')) = h(0.5) = 0.6931$$

The intuition here is that removing context from a difficult image makes the model more uncertain even though it assigns higher probability to the correct class. This models real scenarios where background removal eliminates both distractors and helpful context.

**Verification of the Counterexample.** For the selectively cropped dataset $\mathcal{D}' = \{(x_1, y_1), (x_2', y_2)\}$:

$$\text{NLL}(\mathcal{D}') = \frac{1}{2}[-\ln(0.7) - \ln(0.5)] = \frac{1}{2}[0.3567 + 0.6931] = 0.5249 < 0.9831 = \text{NLL}(\mathcal{D})$$

$$H(\mathcal{D}') = \frac{1}{2}[0.6109 + 0.6931] = 0.6520 > 0.5557 = H(\mathcal{D})$$

Thus we have constructed a concrete example where $\text{NLL}(\mathcal{D}') < \text{NLL}(\mathcal{D})$ yet $H(\mathcal{D}') > H(\mathcal{D})$, proving that lower NLL does not imply lower entropy.

$\square$

## A.2. Proof of Theorem 4.2

**Theorem A.2** (Restated). *Let $\mathcal{D}' = \mathcal{C}_{sel}(\mathcal{D})$ be a selectively cropped dataset with lower initial entropy: $H(\mathcal{D}') < H(\mathcal{D})$. Under Assumption A.3, there exists a crop ratio $r^* \in (0,1)$ such that when random cropping augmentation is applied, the entropy advantage is lost:*

$$H(\mathcal{D}') < H(\mathcal{D}) \quad but \quad H(\mathcal{A}_{r^*}(\mathcal{D}')) \geq H(\mathcal{A}_{r^*}(\mathcal{D})),$$

*where $H(\mathcal{A}_r(\cdot))$ represents the expected entropy over all random spatial crops with ratio $r$.*

**Assumption A.3** (Compounding Information Loss). For the selectively cropped dataset $\mathcal{D}' = \mathcal{C}_{sel}(\mathcal{D})$, there exists a dataset-dependent threshold $r_0 \in (0,1)$ such that for all $r \in (0, r_0)$, aggressive random cropping causes more severe entropy increase than for the original dataset:

$$H(\mathcal{A}_r(\mathcal{D}')) > H(\mathcal{A}_r(\mathcal{D}))$$

*Justification:* This assumption is natural because $\mathcal{D}'$ has already lost spatial context through selective cropping. When an image has been pre-cropped to remove "hard" regions, the remaining content has less redundancy. Further random cropping of this already-reduced image is more likely to remove critical discriminative features, leading to higher prediction uncertainty compared to random cropping of the original, full images. We provide empirical validation in Table 12.

*Proof.* We establish the existence of $r^*$ where the entropy advantage is lost using the Intermediate Value Theorem.

**Setup.** Define the entropy functions for both datasets under random cropping augmentation with parameter $r \in (0,1]$:

$$f(r) := H(\mathcal{A}_r(\mathcal{D})) = \mathbb{E}_{\text{crop} \sim \mathcal{A}_r} \left[ \frac{1}{N} \sum_{i=1}^{N} H(p_{\theta_0}(\cdot|\text{crop}(x_i))) \right] \tag{3}$$

$$f'(r) := H(\mathcal{A}_r(\mathcal{D}')) = \mathbb{E}_{\text{crop} \sim \mathcal{A}_r} \left[ \frac{1}{N} \sum_{i=1}^{N} H(p_{\theta_0}(\cdot|\text{crop}(x_i'))) \right] \tag{4}$$

where the expectation is taken over all possible random crops with area ratio $r$.

**Continuity.** Both $f(r)$ and $f'(r)$ are continuous functions on $(0,1]$. This follows from the composition of continuous operations: random cropping operations employ bilinear interpolation, ensuring continuous transformations as $r$ varies; the neural network $p_{\theta_0}$ consists of continuous activation functions; the entropy function $H(p) = -\sum_y p(y) \ln p(y)$ is continuous in the probability distribution $p$; and the expectation operation preserves continuity when integrated over a continuous parameter space.

**Boundary Analysis.** At $r = 1$ (no effective cropping):

$$f(1) = H(\mathcal{D}) \tag{5}$$
$$f'(1) = H(\mathcal{D}') \tag{6}$$

By hypothesis, $f'(1) = H(\mathcal{D}') < H(\mathcal{D}) = f(1)$.

**Application of the Intermediate Value Theorem.** Define the difference function:

$$g(r) := f'(r) - f(r) = H(\mathcal{A}_r(\mathcal{D}')) - H(\mathcal{A}_r(\mathcal{D}))$$

Having defined the difference function $g(r) := f'(r) - f(r)$, we can now establish its key properties. First, at the boundary $r = 1$, we have $g(1) = H(\mathcal{D}') - H(\mathcal{D}) < 0$ by our initial hypothesis that the selectively cropped dataset has lower entropy. Second, Assumption A.3 guarantees that $g(r) > 0$ for all $r \in (0, r_0)$, reflecting the compounding information loss under aggressive cropping. Finally, since both $f$ and $f'$ are continuous functions on $(0,1]$, their difference $g$ inherits this continuity on the same domain.

Since $g$ is continuous on $[r_0/2, 1]$, with $g(r_0/2) > 0$ and $g(1) < 0$, the Intermediate Value Theorem guarantees the existence of at least one $r^* \in (r_0/2, 1) \subset (0,1)$ such that $g(r^*) = 0$. This establishes:

$$H(\mathcal{A}_{r^*}(\mathcal{D}')) = H(\mathcal{A}_{r^*}(\mathcal{D}))$$

Therefore, there exists $r^* \in (0,1)$ where the initial entropy advantage is completely lost. $\square$

**Empirical Validation of Assumption A.3.** Table 12 validates our compounding assumption by computing $g(r) = H(\mathcal{A}_r(\mathcal{D}')) - H(\mathcal{A}_r(\mathcal{D}))$ across different crop ratios. We model $H(\mathcal{A}_r(\mathcal{D}))$ using single crops with ratio $r$, and $H(\mathcal{A}_r(\mathcal{D}'))$ using consecutive crops (two crops each with ratio $\sqrt{r}$, giving effective ratio $r$), representing the compounding effect of selective cropping followed by random augmentation.

*Table 12.* Validation of Assumption A.3: $g(r) = H(\mathcal{A}_r(\mathcal{D}')) - H(\mathcal{A}_r(\mathcal{D})) > 0$ for small $r$. We model $H(\mathcal{A}_r(\mathcal{D}))$ with single crops and $H(\mathcal{A}_r(\mathcal{D}'))$ with consecutive crops (each with ratio $\sqrt{r}$, total effective ratio $r$).

| Crop Ratio | Quantity | Easy Only | Easy+Bal. | Random | Hard+Bal. | Hard Only |
|---|---|---|---|---|---|---|
| | $H(\mathcal{A}_r(\mathcal{D}))$ | 0.21 | 0.30 | 0.63 | 0.26 | 0.23 |
| $r = 0.08$ | $H(\mathcal{A}_r(\mathcal{D}'))$ | 0.22 | 0.35 | 0.84 | 0.32 | 0.27 |
| | $g(r) = \Delta H$ | +0.01 | +0.05 | +0.21 | +0.06 | +0.04 |
| | $H(\mathcal{A}_r(\mathcal{D}))$ | 0.12 | 0.15 | 0.44 | 0.16 | 0.13 |
| $r = 0.2$ | $H(\mathcal{A}_r(\mathcal{D}'))$ | 0.15 | 0.19 | 0.55 | 0.21 | 0.17 |
| | $g(r) = \Delta H$ | +0.03 | +0.04 | +0.11 | +0.05 | +0.04 |
| | $H(\mathcal{A}_r(\mathcal{D}))$ | 0.09 | 0.08 | 0.21 | 0.10 | 0.09 |
| $r = 0.5$ | $H(\mathcal{A}_r(\mathcal{D}'))$ | 0.13 | 0.12 | 0.28 | 0.15 | 0.12 |
| | $g(r) = \Delta H$ | +0.04 | +0.04 | +0.07 | +0.05 | +0.03 |
| | $H(\mathcal{A}_r(\mathcal{D}))$ | 0.07 | 0.06 | 0.10 | 0.07 | 0.06 |
| $r = 0.8$ | $H(\mathcal{A}_r(\mathcal{D}'))$ | 0.10 | 0.09 | 0.14 | 0.10 | 0.09 |
| | $g(r) = \Delta H$ | +0.03 | +0.03 | +0.04 | +0.03 | +0.03 |

All values of $g(r)$ are positive across all dataset types and crop ratios tested, with the effect most pronounced at $r = 0.08$ where $g(r) = 0.21$ for random datasets, representing a 33% relative increase in entropy. The monotonic decrease of $g(r)$ as $r$ increases is consistent with our theoretical analysis, as the compounding effect diminishes when crops retain more of the original image. This empirical evidence strongly supports Assumption A.3.

**Lemma A.4** (Uncertainty and Generalization). *In small-data regimes and under typical calibration assumptions, datasets exhibiting lower average predictive entropy $H(Y|X; \theta)$ tend to be more trainable and yield better downstream generalization performance. This relationship has been observed in multiple empirical studies (Mukhoti et al., 2020; Coleman et al., 2020).*

**Practical Implications.** The combination of Lemma A.4 and Theorem 4.2 offers a cautionary insight: while selective cropping may reduce entropy during dataset preparation, this advantage is lost (or reversed) under standard training augmentations. In small-data regimes where lower entropy correlates with better generalization, such loss means that models trained on selectively cropped datasets may underperform compared to those trained on uncropped, pruned datasets. This supports our PCA framework's design choice to avoid cropping and instead combine full, pruned images, preserving maximum information and ensuring reliable downstream performance.

# B. Experiment Settings

## B.1. Dataset and Network

**Dataset.** The ImageNet-1K dataset (Deng et al., 2009), also known as ILSVRC-2012, is a large-scale image classification dataset containing $N = 1.28$ million training images and $50,000$ validation images across $K = 1,000$ object categories. Each image is manually annotated with a single class label. The dataset contains approximately $1,200$ images per class in the training set. Images have an average resolution of $469 \times 387$ pixels but are typically pre-processed to a standard size of $224 \times 224$ pixels for model training. This dataset has become a de facto benchmark for evaluating deep learning models in computer vision tasks, particularly for image classification problems.

**Network.** For all networks, we use common network definition from https://pytorch.org/vision/main/models.html. Networks are trained for 300 epochs by default; detailed settings are provided in Appendix B.2.

## B.2. Standard Evaluation Setting

Table 13 provides a more comprehensive comparison among baseline dataset distillation methods. We have adopted the CDA's setting (Yin and Shen, 2024) as the **standard evaluation setting** for two main reasons: (1) many other studies, such as LPLD (Xiao and He, 2024) and DWA (Du et al., 2024), have used this setting; and (2) it applies to most methods, being designed explicitly for datasets that include combined image patterns, in contrast to patch shuffling. Note that baseline dataset pruning methods also adhere to the **standard evaluation setting** for fair comparison.

It's important to note that using alternative settings or additional techniques is **NOT** incorrect; however, we have chosen a common standard evaluation setting to facilitate a clearer comparison among the different methods.

*Table 13.* Inconsistent evaluation settings of Dataset Distillation on ImageNet-1K. Values marked in **bold** are settings different from SRe$^2$L. $^{\dagger}$ represents the IPC-dependent.

| Configuration | Value | SRe$^2$L (Yin et al.) | CDA (Yin and Shen) | LPLD (Xiao and He) | DWA (Du et al.) | RDED (Sun et al.) | G-VBSM (Shao et al.) | EDC (Shao et al.) |
|---|---|---|---|---|---|---|---|---|
| Epochs | 300 | ✓ | ✓ | ✓ | ✓ | ✓ | ✓ | ✓ |
| Optimizer | AdamW | ✓ | ✓ | ✓ | ✓ | ✓ | ✓ | ✓ |
| Model LR | 0.001 | ✓ | ✓ | ✓ | ✓ | ✓ | ✓ | ✓ |
| LR | Smooth LR | ✗ | ✗ | ✗ | ✗ | ✓ | ✗ | ✓ |
| LR Scheduler | CosineAnnealing | ✓ | ✓ | ✓ | ✓ | ✓ | ✓ | ✓ |
| Batch Size | 1024 | 1024 | **128** | **128** | **128** | **100**$^{\dagger}$ | 1024 | **100** |
| Soft Label | Single / Ensemble | Single | Single | Single | Single | Single | **Ensemble** | **Ensemble** |
| Loss Type | KL / MSE+0.1xGT | KL | KL | KL | KL | KL | **MSE** | **MSE** |
| EMA-based | | ✗ | ✗ | ✗ | ✗ | ✗ | ✗ | ✓ |
| | PatchShuffle | ✗ | ✗ | ✗ | ✗ | ✓ | ✗ | ✗ |
| | ResizedCrop | ✓ | ✓ | ✓ | ✓ | ✓ | ✓ | ✓ |
| Augmentation | CropRange | (0.08, 1) | (0.08, 1) | (0.08, 1) | (0.08, 1) | **(0.5, 1)** | (0.08, 1) | **(0.5, 1)** |
| | Flip | ✓ | ✓ | ✓ | ✓ | ✓ | ✓ | ✓ |
| | Cut-Mix | ✓ | ✓ | ✓ | ✓ | ✓ | ✓ | ✓ |

**Remark:** Table 13 does not cover all the different settings. For example, EDC (Shao et al., 2024b) uses EMA-based evaluation while other methods do not include it.

## B.3. Fair Storage of Pruning Datasets

When considering the pruning ratio in state-of-the-art (SOTA) pruning methods, it is important to note that the pruning ratio does not directly correspond to the dataset distillation setting. Existing pruning techniques primarily focus on tracking the ranking of images (i.e., the indices) rather than storing the actual dataset, which leads to the neglect of the true size of the ImageNet-1K images. Additionally, dataset distillation limits image resolution to $224 \times 224$ pixels. Therefore, it is unfair, in terms of information content and storage, to directly store the actual ImageNet-1K images, which have a higher average resolution of $469 \times 387$ pixels. To address this, we choose to crop the images based on their shortest side and then resize them to $224 \times 224$ pixels.

## B.4. Baselines Specifications

In this section, we provide more specifications of each baseline.

**Dataset Distillation Baselines:**

- **SRe$^2$L (Yin et al., 2023)**: No special adjustments. Dataset recovered following https://github.com/VILA-Lab/SRe2L.
- **CDA (Yin and Shen, 2024)**: No special adjustments; results reported are from the original paper. Dataset recovered following https://github.com/VILA-Lab/CDA.
- **G-VBSM (Shao et al., 2024a)**: No special adjustments. Dataset recovered following https://github.com/shaoshitong/G_VBSM_Dataset_Condensation.
- **LPLD (Xiao and He, 2024)**: No special adjustments; results reported are from the original paper. Dataset provided in https://github.com/he-y/soft-label-pruning-for-dataset-distillation.
- **DWA (Du et al., 2024)**: No special adjustments; results reported are from the original paper. Dataset recovered following https://github.com/AngusDujw/Diversity-Driven-Synthesis.
- **RDED (Sun et al., 2024)**: IPC10 and IPC50 selects patch from $m = 300$ patches, and IPC100 selects from $m = 600$ patches. Dataset recovered following https://github.com/LINs-lab/RDED.

**Dataset Pruning Baselines:** We create datasets by using the data ranking scores provided by Zheng et al. (https://github.com/haizhongzheng/Coverage-centric-coreset-selection). After obtaining the ranking, we post-process the datasets into images of resolution $224 \times 224$, according to Appendix B.3.

- **Forgetting (Toneva et al., 2019)**: Images with low "forgetting events" are selected; if images have a same number of "forgetting events", we randomly sample the images. Strict class balance is enforced.
- **EL2N (Paul et al., 2021)**: Images with low "EL2N Scores" are selected; and strict class balance is enforced.
- **AUM (Pleiss et al., 2020)**: Images with high "accumulated margin" are selected; strict class balance is enforced.
- **CCS (Zheng et al., 2023)**: For the base pruning metric, we use AUM (Pleiss et al., 2020) following the original experiment setting. In addition, we prune away 30% "mislabeled" data for IPC10 and IPC50, and 20% "mislabeled" data are removed for IPC100 due to strict class balance requiring enough images for each class.

# C. Results with Standard Deviation

Table 14 provides the standard deviation of the performance of dataset compression methods under the same evaluation setting.

*Table 14.* Benchmarking SOTA methods against random baseline under evaluation with **soft labels** (top) and **hard labels** (bottom) with standard deviation. $^\dagger$ means optimization-free distillation approaches. All experiments use ResNet-18 on ImageNet-1K. Standard deviations are computed from three independent runs.

*(a)* Soft label benchmark with standard deviation.

| IPC | Random | DD (Noise Init) | | | | DD (Real Init) | | Pruning Method with Rules | | | |
| | | SRe$^2$L | CDA | G-VBSM | LPLD | RDED$^\dagger$ | DWA | Forgetting | EL2N | AUM | CCS |
|---|---|---|---|---|---|---|---|---|---|---|---|
| 10 | $35.8_{\pm 0.2}$ | $33.5_{\pm 0.2}$ | $33.5_{\pm 0.3}$ | $35.8_{\pm 0.7}$ | $34.6_{\pm 0.9}$ | $38.4_{\pm 0.1}$ | $37.9_{\pm 0.2}$ | $36.1_{\pm 0.3}$ | $40.8_{\pm 0.4}$ | $41.5_{\pm 0.1}$ | $37.4_{\pm 0.2}$ |
| 50 | $57.2_{\pm 0.2}$ | $52.6_{\pm 0.1}$ | $53.5_{\pm 0.3}$ | $54.8_{\pm 0.2}$ | $55.4_{\pm 0.3}$ | $56.2_{\pm 0.2}$ | $55.2_{\pm 0.2}$ | $57.2_{\pm 0.1}$ | $58.1_{\pm 0.1}$ | $58.5_{\pm 0.1}$ | $58.2_{\pm 0.1}$ |
| 100 | $61.2_{\pm 0.2}$ | $57.4_{\pm 0.3}$ | $58.0_{\pm 0.2}$ | $59.2_{\pm 0.1}$ | $59.4_{\pm 0.2}$ | $60.2_{\pm 0.1}$ | $59.2_{\pm 0.3}$ | $61.0_{\pm 0.1}$ | $61.5_{\pm 0.2}$ | $61.5_{\pm 0.1}$ | $61.6_{\pm 0.1}$ |

*(b)* Hard label benchmark with standard deviation.

| IPC | Random | DD (Noise Init) | | | | DD (Real Init) | | Pruning Method with Rules | | | | PCA |
| | | SRe$^2$L | CDA | G-VBSM | LPLD | RDED$^\dagger$ | DWA | Forgetting | EL2N | AUM | CCS | Ours$^\dagger$ |
|---|---|---|---|---|---|---|---|---|---|---|---|---|
| 10 | $4.6_{\pm 0.1}$ | $1.5_{\pm 0.1}$ | $1.6_{\pm 0.1}$ | $1.6_{\pm 0.1}$ | $3.4_{\pm 0.1}$ | $11.5_{\pm 0.1}$ | $1.9_{\pm 0.0}$ | $3.4_{\pm 0.1}$ | $12.2_{\pm 0.3}$ | $11.4_{\pm 0.0}$ | $6.8_{\pm 0.3}$ | $22.8_{\pm 0.3}$ |
| 50 | $20.6_{\pm 0.1}$ | $3.8_{\pm 0.0}$ | $5.8_{\pm 0.3}$ | $9.0_{\pm 0.6}$ | $5.1_{\pm 0.1}$ | $30.8_{\pm 0.4}$ | $5.3_{\pm 0.2}$ | $11.7_{\pm 0.2}$ | $31.1_{\pm 0.3}$ | $30.6_{\pm 0.1}$ | $29.3_{\pm 0.4}$ | $39.1_{\pm 0.2}$ |
| 100 | $31.7_{\pm 0.6}$ | $4.9_{\pm 0.2}$ | $8.0_{\pm 0.1}$ | $16.6_{\pm 0.6}$ | $6.0_{\pm 0.1}$ | $39.2_{\pm 0.6}$ | $7.5_{\pm 0.1}$ | $18.3_{\pm 0.2}$ | $38.7_{\pm 0.1}$ | $38.8_{\pm 0.2}$ | $39.0_{\pm 0.4}$ | $45.5_{\pm 0.4}$ |

# D. Additional Experiments and Analysis

## D.1. Regularization-based Data Augmentation

*Table 15.* Effects of regularization-based augmentations on PCA (SGD setting). "Crop" refers to *RandomResizedCrop* (0.08-1.00 range). "Mix Probability" indicates the likelihood of applying data mixing, where 1.0 means always applying data mixing. "Label Mixing" combines class labels proportionally to the area of mixed image regions. ResNet-18, IPC10, ImageNet-1K.

*(a)* With RandomResizedCrop

| Crop | Data Mixing | Label Mixing | Mix Probability | | |
|------|-------------|--------------|------|------|------|
| | | | 0.2 | 0.5 | 1.0 |
| ✓ | ✗ | - | | 25.6 | |
| ✓ | CutMix | ✓ | $23.8_{\downarrow 1.8}$ | $23.0_{\downarrow 2.6}$ | $17.4_{\downarrow 8.2}$ |
| | | ✗ | $25.5_{\downarrow 0.1}$ | $24.7_{\downarrow 0.9}$ | $23.0_{\downarrow 2.6}$ |
| ✓ | Mixup | ✓ | $25.7_{\uparrow 0.1}$ | $23.0_{\downarrow 2.6}$ | $7.7_{\downarrow 17.9}$ |
| | | ✗ | $25.9_{\uparrow 0.3}$ | $25.1_{\downarrow 0.5}$ | $17.6_{\downarrow 8.0}$ |
| ✓ | **Cutout** | - | $26.2_{\uparrow 0.6}$ | $25.7_{\uparrow 0.1}$ | $25.3_{\downarrow 0.3}$ |

*(b)* Without RandomResizedCrop

| Crop | Data Mixing | Label Mixing | Mix Probability | | |
|------|-------------|--------------|------|------|------|
| | | | 0.2 | 0.5 | 1.0 |
| ✗ | ✗ | - | | 21.6 | |
| ✗ | CutMix | ✓ | $9.8_{\downarrow 11.8}$ | $8.1_{\downarrow 13.5}$ | $10.5_{\downarrow 11.1}$ |
| | | ✗ | $15.6_{\downarrow 6.0}$ | $14.3_{\downarrow 7.3}$ | $12.5_{\downarrow 9.1}$ |
| ✗ | Mixup | ✓ | $18.9_{\downarrow 2.7}$ | $17.4_{\downarrow 4.2}$ | $8.4_{\downarrow 13.2}$ |
| | | ✗ | $19.2_{\downarrow 2.4}$ | $18.3_{\downarrow 3.3}$ | $15.6_{\downarrow 6.0}$ |
| ✗ | **Cutout** | - | $22.7_{\uparrow 1.1}$ | $22.4_{\uparrow 0.8}$ | $21.8_{\uparrow 0.2}$ |

Table 15 presents a comprehensive evaluation of various data augmentation strategies, including CutMix (Yun et al., 2019), Cutout (DeVries and Taylor, 2017), and Mixup (Zhang, 2017). The experimental results demonstrate the **crucial role of appropriate augmentation selection** in data-scarce scenarios. The incorporation of *RandomResizedCrop* proves to be fundamental, substantially improving performance from 21.6% to 25.6%.

Among the regularization-based augmentation techniques, Cutout demonstrates better performance, maintaining consistent accuracy levels (26.2%, 25.7%, and 25.3% with *RandomResizedCrop*). This superiority can be attributed to two key factors: First, Cutout preserves label integrity by avoiding label mixing, which is particularly beneficial in data-scarce regimes. Second, its augmentations are performed on individual images without cross-sample interactions, adhering to the principle of maintaining sample simplicity during training. In contrast, both CutMix and Mixup show notable performance degradation with increased mixing probabilities, which is especially evident in scenarios with label mixing. When label mixing is employed, performance deteriorates significantly (from 25.5% to 23.8% for CutMix, and from 25.9% to 25.7% for Mixup at 0.2 mixing probability with *RandomResizedCrop*). This degradation becomes more severe at higher mixing probabilities, with performance dropping to 17.4% and 7.7%, respectively, at 1.0 mixing probability.

These findings align with our theoretical framework, suggesting that augmentation strategies maintaining sample simplicity are more effective in data-scarce regimes. The empirical evidence demonstrates that methods introducing complex regularization through label mixing and cross-sample interactions may be detrimental to model performance when training data is limited, supporting our scaling law observations regarding the preference for simpler training samples.

Setting for each strategy:

- CutMix (Yun et al., 2019): We follow the original implementation which samples from $\text{Beta}(\alpha, \alpha)$, where $\alpha = 1$, which is basically uniform sampling from $(0, 1)$. For the label mixing part, we rescale $\lambda$ following https://github.com/clovaai/CutMix-PyTorch.
- Mixup (Zhang, 2017): We follow the original implementation which samples from $\text{Beta}(\alpha, \alpha)$, where $\alpha = 1$, which is basically uniform sampling from $(0, 1)$.
- Cutout (DeVries and Taylor, 2017): We use a common cutout size which is $0.5$.

**Remark:** In the original implementation of SRe²L, CutMix and Mixup do not incorporate label mixing because distillation loss is used.

## D.2. Poor Performance using Forgetting (Toneva et al., 2019)

Figure 6 illustrates the distribution of various score metrics, specifically EL2N (Paul et al., 2021), Forgetting (Toneva et al., 2019), and AUM (Pleiss et al., 2020) Scores. These distributions are organized into two rows, with the top row representing the full dataset and the bottom row depicting the "easiest" IPC10 subset.

In the analysis of the **EL2N Score**, the histogram for the full dataset shows a unimodal distribution that peaks around a score of 10, indicating that most scores are concentrated in this range. Additionally, there is a long tail in the distribution

towards lower scores.

Examining the **Forgetting Score**, the Full dataset displays a bimodal distribution with significant frequencies at scores of 0 and 10. This bimodality indicates the presence of two prevalent score categories within the complete dataset. Conversely, the IPC10 Forget Score distribution is dominated by a sharp peak at score 0, reflecting a substantial proportion of instances with no forgetting behavior in the IPC10 subset.

Regarding the **AUM Score**, the Full dataset illustrates a symmetric distribution centered around a score of 0, indicating balanced score dispersion. The IPC10 AUM Score distribution, however, shows a broader range with a prominent peak near 56 and a gradual decline as scores approach 60. This shift suggests that the IPC10 subset experiences a different range of AUM Scores compared to the full dataset.

The poor performance of forgetting can possibly be explained by the score distribution (see Figure 6). We can clearly see that the easiest IPC10 subsets of forgetting scores all have a value of "0," indicating that no forgetting occurs. Because of the nature of the forgetting approach, many images experience no forgetting events at all. In fact, there are approximately 110,000 images without any forgetting events, and we randomly selected 10,000 (roughly 9.1%) of these images to create our IPC10 dataset. As a result, the 10,000 images are **indistinguishable** from the remaining images (90.9%) that also have zero forgetting counts.

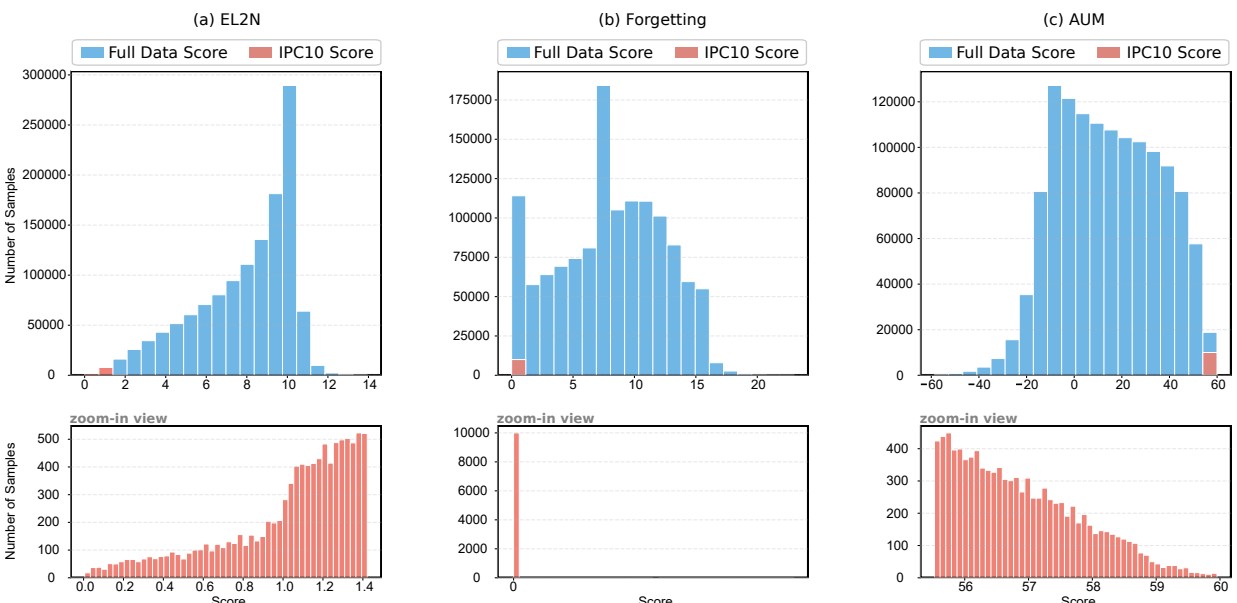

*Figure 6.* Sample distribution over the score of different pruning metrics: (a) EL2N, (b) Forgetting, and (c) AUM. Top row: both the sample distribution of full and IPC10 datasets. Bottom row: zoomed-in view of the distribution of IPC10 dataset. IPC10 datasets are selected from the "easiest" samples.

## D.3. Training with Noisy Images

From Table 16, we can see that even with **purely noisy images**, the student network is able to learn from the teacher network by matching the soft labels. This is surprising, as noisy images are typically not expected to contain any useful information for the network's learning process. Nevertheless, the performance of 0.5% is significant compared to the purely random network's performance of 0.1%.

*Table 16.* Distillation training with pure noise on ResNet-18 on ImageNet-1K.

|       | Expected Acc. | Batch Size =128 | Batch Size =1024 |
|-------|---------------|------------------|-------------------|
| IPC50 | 0.1 %         | 0.5 %            | 0.3 %             |

## D.4. Use Real Images as Initialization for Dataset Distillation

As shown in Figure 3a, we categorize existing literature into three distinct sections. The first section encompasses dataset distillation with **noise** initialization, where no images from the original dataset are directly involved. The representative work in this category is SRe$^2$L (Yin et al., 2023), which pioneered this approach. The second section comprises dataset distillation with **real** image initialization, where the number of original images directly involved equals the distilled dataset size (specifically, IPC $\times$ 1000 images). An exception is RDED (Sun et al., 2024), which randomly samples $m$ images and combines crops, utilizing $m \times 1000$ images, where $m >$ IPC. The final section focuses on dataset pruning methods, which evaluate the entire dataset to identify optimal subsets, thereby involving all images directly in the dataset compression process.

To validate the significance of incorporating more original images, we reimplemented SRe$^2$L with real images as initialization. Table 17 demonstrates that merely initializing with real images consistently improves performance across both soft-label and hard-label benchmarks.

*Table 17.* Performance of SRe$^2$L with real images as initialization.

| | Soft Label | | | Hard Label | | |
|---|---|---|---|---|---|---|
| | Random | SRe$^2$L | SRe$^2$L$_{Real}$ | Random | SRe$^2$L | SRe$^2$L$_{Real}$ |
| 10 | $35.8_{\pm 0.2}$ | $33.5_{\pm 0.2}$ | $35.3_{\pm 0.5}$ | $4.6_{\pm 0.1}$ | $1.5_{\pm 0.1}$ | $2.5_{\pm 0.0}$ |
| 50 | $57.2_{\pm 0.2}$ | $52.6_{\pm 0.1}$ | $53.9_{\pm 0.3}$ | $20.6_{\pm 0.1}$ | $3.8_{\pm 0.0}$ | $6.3_{\pm 0.2}$ |
| 100 | $61.2_{\pm 0.2}$ | $57.4_{\pm 0.3}$ | $58.3_{\pm 0.1}$ | $31.7_{\pm 0.6}$ | $4.9_{\pm 0.2}$ | $7.9_{\pm 0.2}$ |

## D.5. Random Baseline in Soft Label Dataset Distillation

Many soft-label-DD methods (Yin et al., 2023; Yin and Shen, 2024; Du et al., 2024) overlook the random baseline, and we find, when equipped with soft labels, random baselines can attain a good performance. In addition, we provide the random baselines with most common batch sizes, and we advocate that random baselines should be included for comparison in future works.

*Table 18.* Random Baseline with Soft Label Distillation.

| | ResNet-18 | | | ResNet-50 | | ResNet-101 |
|---|---|---|---|---|---|---|
| IPC/BS | 32 | 128 | 1024 | 32 | 128 | 128 |
| 1 | $4.1_{\pm 0.2}$ | $4.3_{\pm 0.1}$ | $1.9_{\pm 0.1}$ | $3.7_{\pm 0.2}$ | $3.6_{\pm 0.1}$ | $3.1_{\pm 0.5}$ |
| 10 | $37.7_{\pm 0.4}$ | $35.8_{\pm 0.2}$ | $23.6_{\pm 0.3}$ | $42.9_{\pm 0.6}$ | $39.3_{\pm 1.6}$ | $37.7_{\pm 1.3}$ |
| 20 | $49.6_{\pm 0.7}$ | $48.5_{\pm 0.1}$ | $38.2_{\pm 0.3}$ | $54.8_{\pm 0.6}$ | $55.5_{\pm 0.2}$ | $52.9_{\pm 3.0}$ |
| 50 | $58.0_{\pm 0.1}$ | $57.2_{\pm 0.2}$ | $52.4_{\pm 0.2}$ | $64.3_{\pm 0.2}$ | $64.2_{\pm 0.1}$ | $62.1_{\pm 2.2}$ |
| 100 | $61.5_{\pm 0.1}$ | $61.2_{\pm 0.2}$ | $58.3_{\pm 0.0}$ | $67.4_{\pm 0.1}$ | $67.0_{\pm 0.2}$ | $65.8_{\pm 0.9}$ |
| 200 | $64.9_{\pm 0.5}$ | $64.2_{\pm 0.1}$ | $61.6_{\pm 0.0}$ | $68.6_{\pm 0.2}$ | $68.8_{\pm 0.1}$ | $69.1_{\pm 0.1}$ |

## D.6. Strict Data Balance Is an Implicit Stratification

Figure 7 (Top) illustrates the distribution of samples across different classes. A clear severe class imbalance is observed when samples are selected solely based on pruning scores, as shown by the red histogram. Some classes have no images at all, while others contain more than 100 images. This imbalance is particularly noticeable when using Forgetting as the pruning metric.

By enforcing strict class balance, the difficulty of the subset increases as long as class imbalance persists. This is demonstrated in Figure 7 (Bottom), where higher scores in EL2N and Forgetting indicate a harder dataset, while a lower score in AUM suggests the opposite. Consequently, strict class balance implicitly achieves data stratification (Zheng et al., 2023) among easy samples, and it can partly explain Table 19 why adding additional explicit stratification does not improve the performance as suggested by CCS (Zheng et al., 2023). Additional stratification applied after strict balancing increases dataset complexity, with particularly noticeable effects in small IPC scenarios.

*Table 19.* CCS performance comparison on soft and hard label settings. CCS$_{\text{AUM}}$ denotes stratification performed on AUM.

| Setting | IPC | Random | Forgetting | AUM | EL2N | CCS$_{\text{AUM}}$ |
|---------|-----|--------|------------|-----|------|--------------------|
| Soft | 10 | $35.8_{\pm 0.2}$ | $36.1_{\uparrow 0.3}$ | $\mathbf{41.5}_{\uparrow 5.7}$ | $40.8_{\uparrow 5.0}$ | $37.4_{\uparrow 1.6}$ |
| | 50 | $57.2_{\pm 0.2}$ | $57.2_{=0.0}$ | $\mathbf{58.5}_{\uparrow 1.3}$ | $58.1_{\uparrow 0.9}$ | $58.2_{\uparrow 1.0}$ |
| | 100 | $61.2_{\pm 0.2}$ | $61.0_{\downarrow 0.2}$ | $61.5_{\uparrow 0.3}$ | $61.5_{\uparrow 0.3}$ | $\mathbf{61.6}_{\uparrow 0.4}$ |
| Hard | 10 | $4.6_{\pm 0.1}$ | $3.4_{\downarrow 1.2}$ | $11.4_{\uparrow 6.8}$ | $\mathbf{12.2}_{\uparrow 7.6}$ | $6.8_{\uparrow 2.2}$ |
| | 50 | $20.6_{\pm 0.1}$ | $11.7_{\downarrow 8.9}$ | $30.6_{\uparrow 10.0}$ | $\mathbf{31.1}_{\uparrow 10.5}$ | $29.3_{\uparrow 8.7}$ |
| | 100 | $31.7_{\pm 0.6}$ | $18.3_{\downarrow 13.4}$ | $38.7_{\uparrow 7.0}$ | $38.8_{\uparrow 7.1}$ | $\mathbf{39.0}_{\uparrow 7.3}$ |

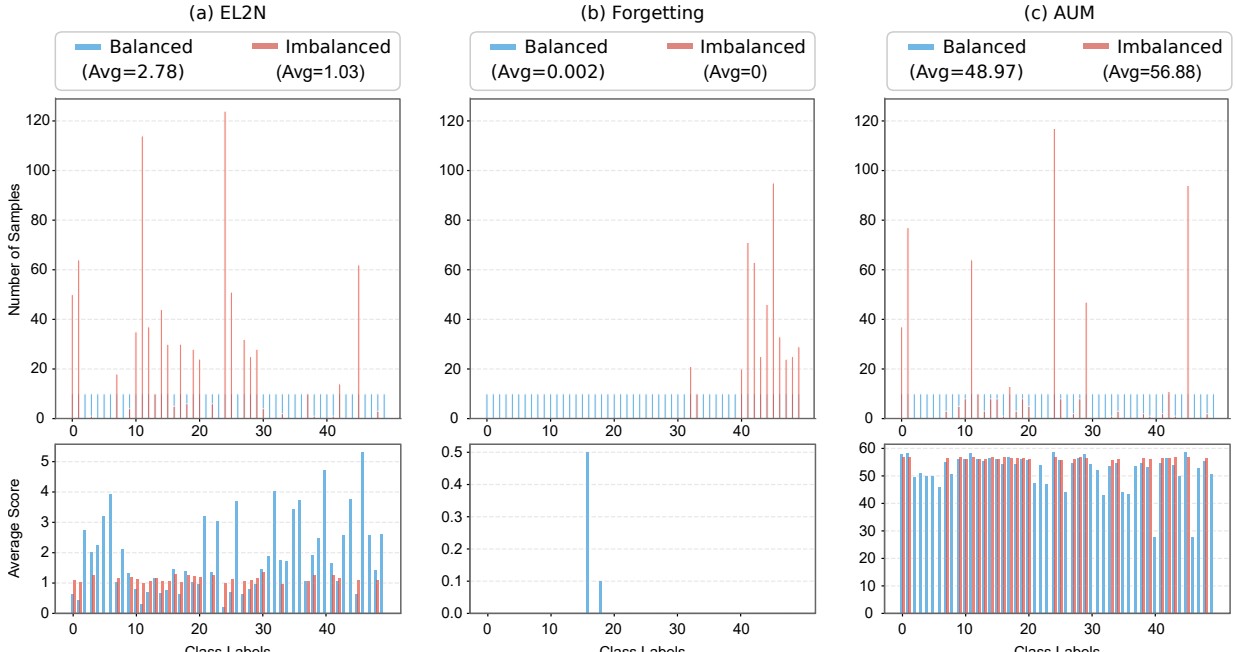

*Figure 7.* Sample and score distribution over class on both balanced and imbalanced cases. Top row: the sample distribution. Bottom row: the score distribution over class. For visualization purposes, only the first 50 classes are presented. IPC10 datasets are visualized and are selected from the "easiest" samples.

## D.7. Difference between Mosaic Augmentation

One approach similar to the "combining" process is Mosaic Augmentation, introduced in YOLOv4 (Bochkovskiy et al., 2020) for object detection tasks, as shown in Figure 8. However, the motivation behind it differs significantly. Combining images consolidates information from multiple sources into a single composite image, thereby saving storage space. In contrast, Mosaic Augmentation mixes multiple (i.e., four) images to facilitate the detection of objects outside their normal context. Additionally, at the implementation level, Mosaic Augmentation loads four times as many images per given batch size, necessitating four times the storage. Nevertheless, the non-uniform combination method could potentially be leveraged in our "combining" approach, which we leave for future study.

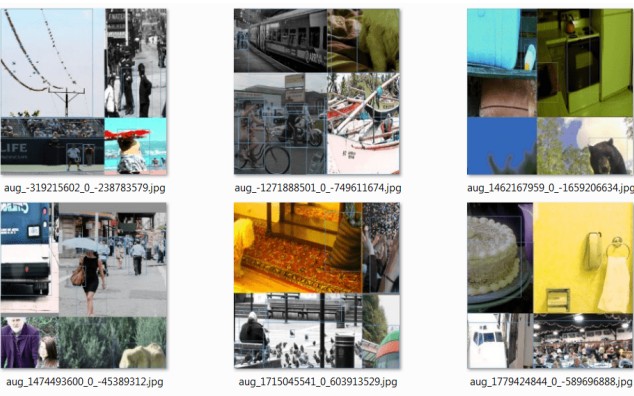

*Figure 8.* Mosaic Augmentation. (Image directly taken from YOLOv4 (Bochkovskiy et al., 2020))

## D.8. Computation Cost and Wall-clock Overhead Analysis

One significant advantage of our PCA framework is its efficiency. Table 20 compares the costs associated with the traditional dataset compression framework, SRe²L, and our PCA method. Among the three stages of SRe²L, the "squeeze" stage is the most time-consuming, particularly when applied to ResNet with the entire ImageNet-1K dataset, which is quite resource-intensive. The parameter storage is 0.04 GB (44M). The second most time-consuming process is the "recover" stage. In contrast, the "relabel" process takes the least amount of time; however, it can become lengthy if the IPC is large due to the introduction of extensive labels, as noted by Xiao and He.

To further illustrate the computational advantages of pruning-based methods over distillation-based approaches, Table 21 presents a comprehensive comparison of wall-clock time across different method categories. The table clearly demonstrates that dataset pruning methods require significantly less computational time compared to dataset distillation methods. In general, pruning methods are much faster, as they require no optimization over images. In addition, these dataset distillation methods optimize for a fixed IPC, and must be rerun for different IPC values. Notably, distillation methods not only require extensive GPU time for image generation but also incur substantial CPU overhead for soft label generation. A detailed breakdown of the relabel timing is provided in Table 22, which shows that the device-to-host transfer time (i.e., moving soft labels from GPU to CPU) significantly contributes to the overall CPU time. This indicates the soft label generation process is CPU-heavy with significant memory-transfer bottlenecks, and such a case can be problematic for devices with limited CPU resources.

On the contrary, let us consider EL2N (Paul et al., 2021), which serves as an example in our primary experiments. The time of the "prune" process involves acquiring the training dynamics, which can be considerably shorter than training the entire model. Furthermore, since our approach is optimization-free, there are no additional costs incurred for combining the images, and we exclusively utilize hard labels instead of soft labels.

*Table 20.* Computation Cost of Dataset Compression between Traditional Framework and PCA. IPC-10, ImageNet-1K.

| SRe²L | Squeeze | Recover | Relabel | | PCA | Prune | Combine |
|---|---|---|---|---|---|---|---|
| Time[1] | 90 epochs | 580 mins | 33 mins | | Time | 10 epochs | - |
| Storage (GB) | 0.04 | 0.15 | 5.67 | | Storage (GB) | - | 0.15 |

---

[1] All time data have been tested on a single RTX30390 GPU card.

*Table 21.* Wall-clock Time Comparison between Dataset Pruning and Dataset Distillation Methods. ImageNet-1K, IPC50.

| Type | Method | Image (GPU-heavy) | Soft Labels (CPU-heavy) |
|---|---|---|---|
| Dataset Pruning | EL2N (Paul et al., 2021) | 1.4 Hrs | - |
| | DUAL (Cho et al., 2025) | 8.5 Hrs | - |
| Dataset Distillation | G-VBSM (Shao et al., 2024a) | 212 Hrs | 2 Hrs |
| | DELT (Shen et al., 2024) | 39 Hrs | 2 Hrs |

*Table 22.* Relabel Cost Breakdown

| Operation | CPU | | GPU | |
|---|---|---|---|---|
| | Time (ms) | Memory (MB) | Time (ms) | Memory (MB) |
| Move Data to GPU | 3.36 | 0.00 | 3.26 | 86717.81 |
| Mix Augmentation | 0.44 | 1.14 | 0.12 | 0.00 |
| Model Inference | 4.42 | 0.00 | 25.33 | 0.46 |
| Move Soft Label to CPU | 22.04 | -0.68 | 0.01 | 0.00 |
| Write Soft Label to Disk | 0.72 | 0.00 | 0.00 | 0.00 |
| Others (83 ops) | 0.95 | 10.22 | 0.87 | 164057.58 |

## D.9. Comparison with RDED

### D.9.1. METHODOLOGICAL INNOVATIONS

While RDED (Sun et al., 2024) serves as a notable baseline in dataset distillation, PCA introduces fundamental improvements that extend beyond marginal patch selection enhancements. Table 23 delineates the key distinctions across three critical stages of the distillation pipeline.

A significant divergence occurs in image preparation, where RDED's random cropping approach inherently fragments the dataset through multiple patches per image, resulting in substantial information loss. In contrast, PCA employs full dataset pruning combined with strategic image scaling, thereby preserving global contextual information throughout the distillation process. Additionally, PCA eliminates the computational burden associated with soft label generation, which is a requirement in RDED that necessitates teacher model dependencies and incurs considerable storage overhead. Although constrained augmentation naturally complements collage-based methods with one-hot labels, its implementation within PCA represents a systematic optimization specifically tailored for small-scale datasets, contrasting sharply with RDED's conventional augmentation strategy that lacks such targeted specialization.

*Table 23.* Comprehensive comparison between RDED and PCA methodologies.

| Stage | RDED (Sun et al., 2024) | PCA (Ours) |
|---|---|---|
| **Image Preparation** | 1. Random subset (300 images) | 1. Full dataset pruning |
| | 2. 5 random crops per image | 2. Image combination by scaling |
| | 3. Patch selection | |
| | 4. Patch combination | |
| | $\rightarrow$ Information loss | $\rightarrow$ Maintains global information |
| **Soft Label Generation** | 1. Requires relabeling process | Not required |
| | 2. Relies on teacher model | |
| | $\rightarrow$ High storage and compute cost | $\rightarrow$ No dependency on teacher models |
| **Dataset Training** | Traditional augmentation | Constrained augmentation |
| | $\rightarrow$ No special emphasis | $\rightarrow$ Optimized for small datasets |

### D.9.2. TRUE CONTRIBUTION OF IMAGES

To provide a comprehensive and fair comparison with RDED, we conducted experiments applying our constrained augmentation strategy to both methods under identical conditions. As shown in Table 24, we evaluate RDED in its original form, with shuffle augmentation, and with our proposed constrained augmentation strategy, comparing these against our PCA framework that also employs constrained augmentation. The results demonstrate that while both methods benefit from the constrained augmentation approach, with RDED improving from 11.4 to 19.2 at IPC10, our PCA framework consistently maintains superior performance across all settings. Specifically, even when both methods utilize identical augmentation strategies, PCA outperforms RDED with constrained augmentation by 6.4, 4.4, and 4.4 percentage at IPC10, IPC50, and IPC100 respectively. This consistent improvement, highlighted in the last row, validates that the performance gains stem from our core methodological innovations rather than merely the augmentation strategy.

*Table 24.* Performance comparison across different methods and IPC settings

| Method | IPC10 | IPC50 | IPC100 |
|---|---|---|---|
| RDED | 11.4 | 30.6 | 39.8 |
| RDED + Shuffle | 12.9 | 32.8 | 41.4 |
| RDED + Constrained Aug | 19.2 | 37.7 | 44.2 |
| Our PCA (with Constrained Aug) | **25.6** | **42.1** | **48.6** |
| True Contribution of Images | ↑6.4 | ↑4.4 | ↑4.4 |

## E. Compute Resources

Experiments of small batch sizes (e.g., batch size 32, 128) are conducted on RTX3090 GPU cards. Experiments of large batch sizes (e.g., batch size 1024) and large networks (e.g., ResNet-50, ResNet-101, SwinV2-Tiny) are conducted on A100 80GB GPU cards.

# F. Visualization

## F.1. Visualization of Dataset Distillation Methods

Figure 9 visualizes the result of SRe$^2$L (Yin et al., 2023). Figure 10 visualizes the result of CDA (Yin and Shen, 2024). Figure 11 visualizes the result of G-VBSM (Shao et al., 2024a). Figure 12 visualizes the result of LPLD (Xiao and He, 2024). Figure 13 visualizes the result of DWA (Du et al., 2024). Figure 14 visualizes the result of RDED (Sun et al., 2024). For all distillation methods (except for RDED (Sun et al., 2024)), images undergo strong distortion.

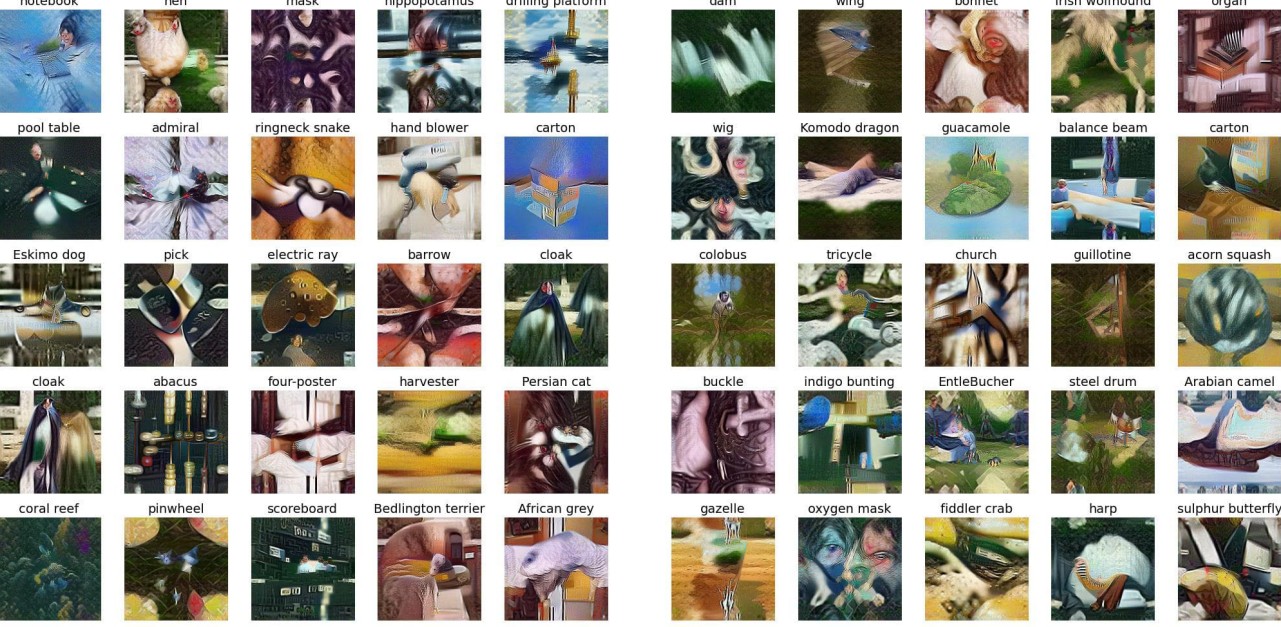

*Figure 9.* SRe$^2$L (Yin et al., 2023)

*Figure 10.* CDA (Yin and Shen, 2024)

*Figure 11.* G-VBSM (Shao et al., 2024a)

*Figure 12.* LPLD (Xiao and He, 2024)

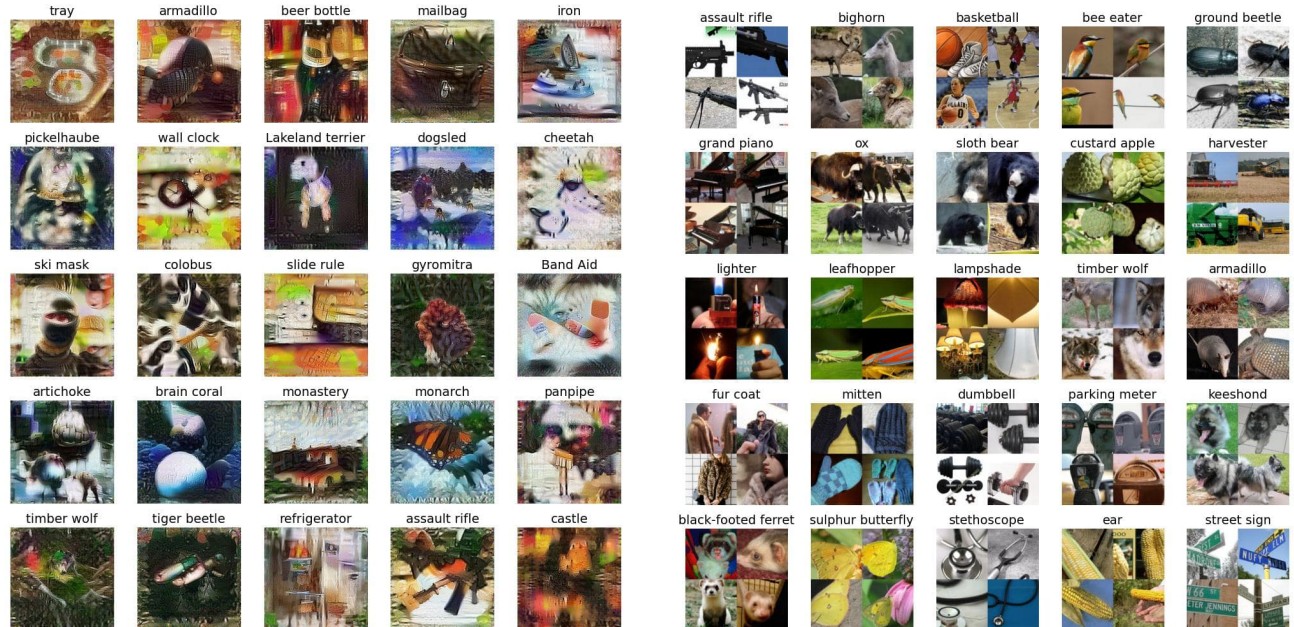

*Figure 13.* DWA (Du et al., 2024)           *Figure 14.* RDED (Sun et al., 2024)

## F.2. Visualization of Dataset Pruning Methods

Figure 15 visualizes the result of Forgetting (Toneva et al., 2019). Figure 16 visualizes the result of AUM (Pleiss et al., 2020). Figure 17 visualizes the result of EL2N (Paul et al., 2021). Figure 18 visualizes the result of CCS (Zheng et al., 2023). The visualization results of all pruning methods followed the pruning rules, allowing for the clear observation that most of the selected images are distinct and visually easy to identify.

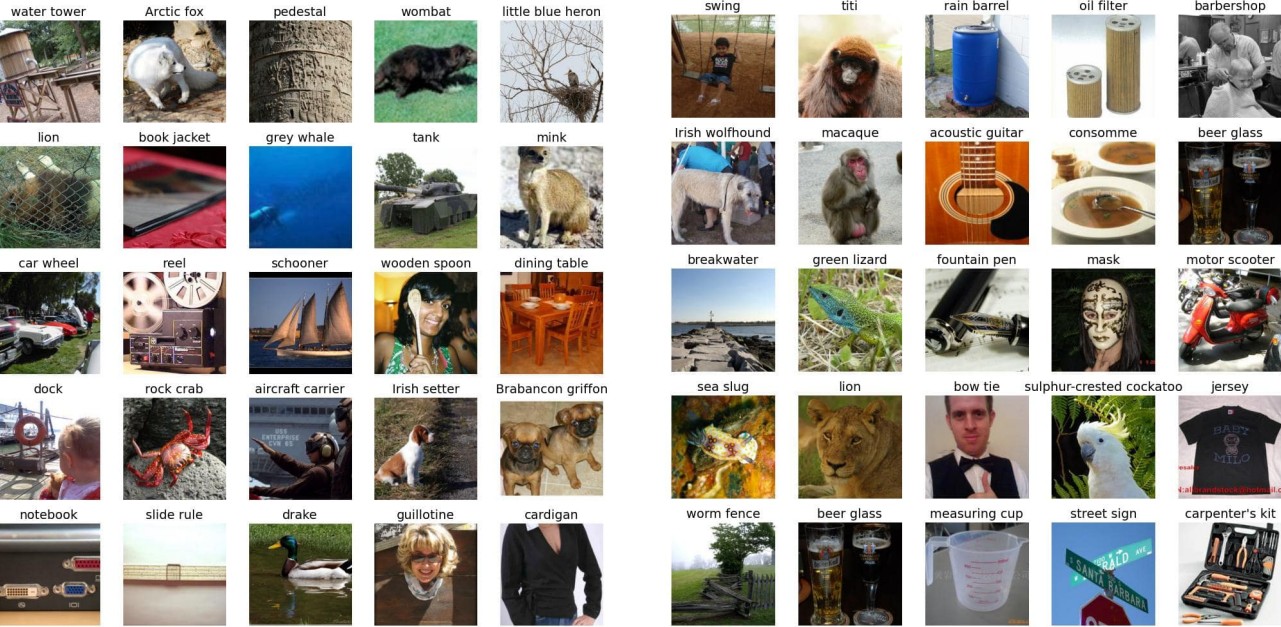

*Figure 15.* Forgetting (Toneva et al., 2019)

*Figure 16.* AUM (Pleiss et al., 2020)

*Figure 17.* EL2N (Paul et al., 2021)

*Figure 18.* CCS (Zheng et al., 2023)

## F.3. Visualization of PCA

Figure 19 shows the images of our PCA framework which uses EL2N (Paul et al., 2021) as the selection metric. Even when adhering to pruning rules, the combined images may not appear visually similar. For example, the "sax" class (first row, second column) demonstrates distinct contexts (i.e., placing the sax on a purple background or a musician playing the sax). This further demonstrates the importance of scaling-law aware augmentation, as inappropriate subsequent training augmentations can lead to a significant difficulty increase in the images.

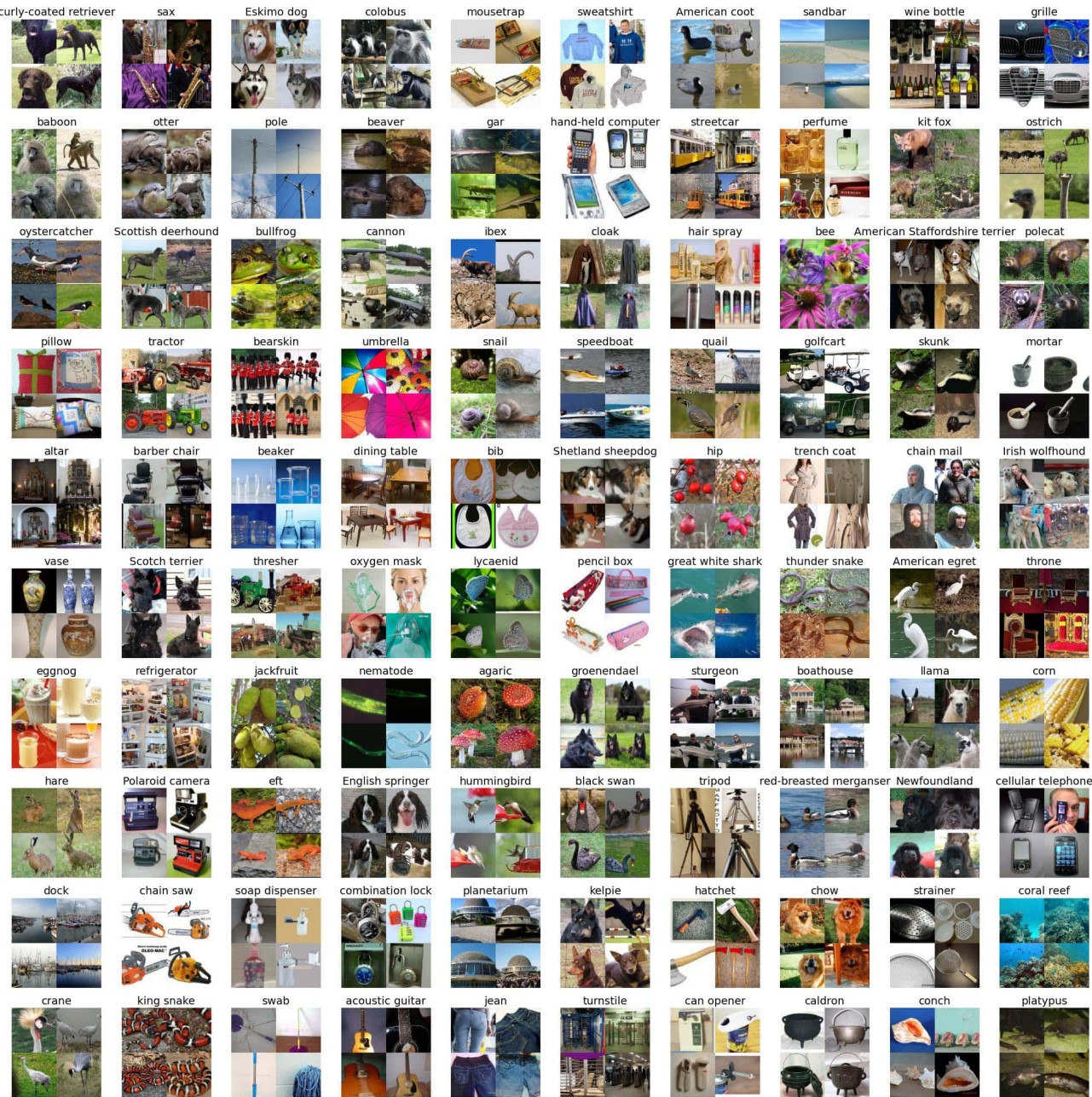

*Figure 19.* Ours (PCA based on EL2N).

## G. Limitation

Our augmentation procedures, including patch extraction, are heuristically designed. While they demonstrate strong empirical effectiveness, their optimality is not theoretically guaranteed.

## H. Future Work

Given that the proposed PCA functions as a framework, there is potential to explore **different choices** of the modules, such as pruning metrics, combining strategies, and specific augmentation methods. It is notable that pruning can extend beyond the original dataset. Instead of only developing new pruning metrics, one could target different datasets. In this paper, the primary reason for pruning on the original dataset is that most existing dataset distillation methods do not outperform random baselines, indicating that original images are sufficiently effective. Hence, there is **significant value** in considering pruning on potentially high-performing distilled datasets (e.g., YOCO (He et al., 2024)) or on generated datasets (e.g., diffusion-based DD methods (Su et al., 2024)). Beyond accuracy, future frameworks might also jointly optimize additional metrics, such as robustness, fairness, or interpretability, while maintaining the same compressed dataset constraint.

