# OpenReview forum: "Unifying Dataset Pruning and Distillation for Efficient Large-scale Compression"
_ICML.cc/2026/Conference — ICML 2026 regular_

### Official Review · Reviewer_9hka · 2026-02-15

**Soundness:** 3
**Presentation:** 3
**Significance:** 3
**Originality:** 3
**Overall Recommendation:** 4
**Confidence:** 4

**Summary:**

The paper investigates the convergence between Dataset Pruning  and Dataset Distillation, proposing a unified Dataset Compression benchmark. The authors highlight a critical paradox: when equipped with soft labels (which consume significant storage), random subsets often outperform complex DD methods. Overall, the submission addresses an important area by challenging the current reliance on soft labels in dataset distillation.

**Compliance With Llm Reviewing Policy:**

Affirmed.

**Key Questions For Authors:**

1. Missing Benchmarks: Could you provide results on CIFAR-10 and CIFAR-100? The DD community relies heavily on these datasets.
2. Clarification on "Fixed Epochs vs. Varying Evaluation": regarding your claim in Section 3 ("DD and DP's Difference 2"):
- Could you explicitly detail what you mean by "varying evaluation settings" in DD? Are you referring to the specific learning rates/optimizers used during the student training phase?
- Why is standardizing DP training (fixed epochs) considered "fair" for DD images which might require different convergence dynamics?

**Limitations:**

The authors primarily discuss limitations in Appendix G, mentioning that their augmentation procedures are heuristic. However, a major limitation not fully addressed is the potential dependency of the "Combine" strategy on high-resolution datasets (like ImageNet). The method might struggle with low-resolution datasets (CIFAR).

**Strengths And Weaknesses:**

Strengths:

1. Critical Insight on Soft Labels
2. Unified Benchmark: The field suffers from inconsistent evaluation settings, and this work attempts to level the playing field.
3. The proposed PCA framework completely eliminates the need for soft labels, significantly reducing storage requirements (up to 40x less than soft-label approaches) while maintaining competitive accuracy on ImageNet-1K.

Weaknesses:

1. Insufficient Experiments: The authors claim "Extensive experiments," yet the evaluation is heavily concentrated on ImageNet-1K. Standard benchmarks in the DD literature, such as CIFAR-10, CIFAR-100, and Tiny-ImageNet, are noticeably missing from the main results.

2. The paper states: "DD methods train a fixed number of epochs regardless of the dataset size" and "DD methods themselves have varying evaluation settings." This argument needs more concrete clarification. For instance, DP usually refers to training on the subset for standard epochs, while DD often involves distinct "synthesis steps" vs. "evaluation steps." Are you comparing the synthesis cost or the final model training cost? Need more specific setting report.

3. The individual components of PCA (using EL2N for pruning, combining patches and constrained augmentation) are not entirely new. The contribution lies in the combination, but the technical depth of the "Combine" strategy seems relatively straightforward (simple collage).

---

> ### Author Rebuttal · Authors · 2026-03-30
>
> Thank you for the positive feedback and the insightful questions. We appreciate your recognition of our contributions and would like to address your concerns one by one.
>
> > W1/Q1/Limitation: Could you provide results on CIFAR-10 and CIFAR-100?
>
> Thank you for the suggestion. We clarify that our claim of "first hard-label dataset compression framework" refers specifically to the **large-scale setting**.
> 1. **Large-scale DD has a unique problem that motivates PCA.** Small-scale DD methods (CIFAR-level) rely on fine-grained matching objectives (gradients, trajectories, etc.) that are computationally prohibitive at ImageNet scale. To bridge this gap, SRe2L introduced the squeeze-recover-relabel paradigm, but at the cost of soft labels with **40x storage overhead**. PCA is specifically designed to eliminate this overhead in large-scale scenarios where such storage constraints become critical.
>
> 2. **The unified DD/DP benchmark is most meaningful at large scale.** The convergence trend we identify, namely DD methods increasingly relying on real images, is primarily a large-scale phenomenon driven by scalability constraints (Sec 1). At CIFAR scale, matching-based DD methods remain effective and the storage overhead of soft labels is less prohibitive (or not existing), so the motivation for hard-label-only compression is less pressing.
>
> 3. **Practical relevance.** The real-world impact of dataset compression lies in large-scale applications where storage and compute constraints are most critical. We acknowledge that additional CIFAR results could provide broader validation, and we will consider including them in the revision as supplementary experiments.
>
>
>
> > Q2: Clarification on "Fixed Epochs vs. Varying Evaluation": regarding your claim in Section 3 ("DD and DP's Difference 2").
>
> Thank you for bringing up the question. We would like to clarify:
>
> 1. **Fixed Iterations and Fixed Epochs**: In DP settings, different pruned datasets (e.g., 50% or 90%) usually share the same training iterations. Here are two specific settings in DP:
> ```
> CCS [a]
> python train.py --dataset cifar10 --gpuid 0 --iterations **40000** --task-name ccs-0.1 ...
>
> DUAL [b]
> python train_imagenet.py --iterations **300000** --iterations-per-testing 5000 ...
> ```
>
> The training process is controlled by a fixed number of iterations, regardless of dataset size. In contrast, DD conventionally trains with 300 epochs, meaning larger datasets naturally undergo more iterations. This discrepancy makes direct comparisons across different dataset sizes potentially unfair.
>
> 2. **Varying Evaluation Settings among DDs (Table 13 in Appendix B.2)**: When evaluating a distilled dataset, a model must be trained on it, and inconsistent training configurations (e.g., batch size, learning rate scheduler, loss function) can obscure the dataset's true contribution, making fair comparisons across different methods unreliable.
>
> [a] Coverage-centric coreset selection for high pruning rates, ICLR'23
>
> [b] Lightweight Dataset Pruning without Full Training via Example Difficulty and Prediction Uncertainty, ICML'25
>
> > Q3: The technical depth of the "Combine" strategy seems relatively straightforward (simple collage).
>
> The depth lies in the theoretical justification rather than procedural complexity.
>
> - Theorem 4.2 provides the proof that "simple collage" (scaling and stitching) is superior to "selective cropping" because it avoids the compounding information loss that occurs during training-time augmentation.
> - By proving that NLL reduction does not guarantee entropy reduction, we justify why a "straightforward" combination of full images is more effective for model training than complex optimization-based synthesis.
>
> We have summarized our contributions in the following tables:
>
> **Table C1: Pruning (Sec 4.1)** *E=Empirical, T=Theoretical*
>
> | Question | Findings | Type | Ref |
> |----------|---------|------|-----|
> | Why use? | Theoretical advantage | T | Fig 4, Lem A.4 |
> | | Performance | E | Fig 3, Tab 2 |
> | Why it works? | Implicit stratification | E | App D.6 |
> | Which method? | EL2N best trade-off | E | Tab 3, App D.2 |
> | | Forgetting limits | E | App D.2 |
> | | Other compatibility | E | Tab 10 |
>
> **Table C2: Combine (Sec 4.2)** *E=Empirical, T=Theoretical*
>
> | Question | Findings | Type | Ref |
> |----------|---------|------|-----|
> | Why cropping-free? | NLL vs. entropy decoupling | T | Prop 4.1 |
> | | Compounding loss under augmentation | T | Thm 4.2 |
> | | Dataset crop impact | E | Tab 8 |
> | | Combination factor ablation | E | Tab A4 (See Reveiwer jfCU Q4) |
>
> **Table C3: Augmentation (Sec 4.3)** *E=Empirical, T=Theoretical*
>
> | Question | Findings | Type | Ref |
> |----------|---------|------|-----|
> | Why constrained? | Compounding loss under training crop | T | Thm 4.2 |
> | | Data-scaling law | E | Tab 5 |
> | | Training crop ratio impact | E | Tab 9 |
> | Other options? | Regularization-based | E | Tab 15 |

---

> > ### Author Rebuttal · Reviewer_9hka · 2026-04-03
> >
> > Thanks for the clarifications, they answered all questions I have. I continue to recommend acceptance.

---

> > > ### Author Response · Authors · 2026-04-03
> > >
> > > Dear Reviewer 9hka,
> > >
> > > We are very encouraged to see that our clarifications have fully resolved your concerns.
> > >
> > > We would be grateful if you could consider reflecting this positive evaluation in your final rating. Thank you for the insightful discussion.

---

### Official Review · Reviewer_eDsu · 2026-03-07

**Soundness:** 3
**Presentation:** 3
**Significance:** 3
**Originality:** 4
**Overall Recommendation:** 5
**Confidence:** 3

**Summary:**

The authors study the dual problems of dataset pruning (DP) and dataset distillation (DD) for creating image datasets, attempting to view them through a unified lens of dataset compression (DC). They first benchmark DP and DC methods by placing them on a sliding scale of (% real images) vs (% distilled/synthesized images) output by the method, and including another axis for hard vs soft labels. They show that soft labels are the main driver of performance advantages of DD methods vs DP methods, yet they come with significant memory and infrastructure costs. To close this gap, they propose the Prune, Combine and Augment (PCA) method for dataset compression which only uses hard labels. Experiments demonstrate that this PCA method outperforms both DP and DD methods in this setting.

**Compliance With Llm Reviewing Policy:**

Affirmed.

**Final Justification:**

My initial review was positive and the rebuttal answered remaining questions satisfactorily, so I continue to recommend acceptance.

**Key Questions For Authors:**

1. How does the "Combine" step of this PCA method work? How are the images to be combined chosen?
2. Does the PCA method generalize to different vision datasets and tasks? Even different visual (or not) modalities?
3. What happens with transformer backbones? It is noted in the paper that transformer backbones are extremely data-hungry; does this mean that you predict in general that this PCA method will not be effective for transformers? Will it at least fare better than competitors?

**Limitations:**

yes

**Strengths And Weaknesses:**

**Strengths:**
- The motivation for the new PCA method is well-explained and defended with both intuition and experiments, making it clear why constraints require the development of a new hard-label-only method.
- The actual PCA method is simple and relatively easy to understand at a high level.
- The analysis is very thorough in terms of past methods surveyed; not just the motivating benchmark but also all subsequent evaluations of the PCA method compare against a breadth of past work.
- The empirical results are good, showing improvement over competitor methods when only hard labels are allowed.

**Weaknesses:**
- Some concrete parts of the PCA framework may not be concretely defined. For example there is not much information about the "Combining" step --- how does the pipeline choose which images to combine and downsample? It would be great to write detailed pseudocode or actual PyTorch (or failing that, a detailed algorithm specification).
- All methods are evaluated only on a single dataset, ImageNet-1K; would be interesting to see if this method could generalize to different datasets (or even modalities).
- All  experiments are done on ResNet and SGD, except for a single experiment in the last row of Table 6 which uses a transformer backbone at a single model scale. It would improve the applicability of the framework to show that it improves on ViTs and Adam optimizer (say)...

---

> ### Author Rebuttal · Authors · 2026-03-30
>
> Thank you for the high score and insightful questions. We would like to address your concerns one by one.
>
> > W1/Q1: How does the "Combine" step of this PCA method work? How are the images to be combined chosen?
>
> The "Combine" step is designed to maximize information retention:
> 1. Selection: We select the "easiest" images based on the reversed EL2N metric applied to the full dataset.
> 2. Shortest-Side Cropping: We perform shortest-side cropping to ensure a fixed 224x224 image resolution, therefore not introducing any potential advatange over dataset distillation approach.
> 3. Downsampling: We use bilinear interpolation to downsample (the default method in pytorch resize function).
> 4. Combination: These downsampled 4 images are stitched by image ids into a single $224 \times 224$ composite.
>
> The pesudocode is provided below:
> ```
> 1. Obtain any image ranking with a pruing method.
> 2. Perform image transformation to a fixed 224x224
>     transform=transforms.Compose([
>         ShortestSideCrop(),
>         transforms.Resize((224, 224)),
>         transforms.ToTensor(),
>     ]),
> 3. Combine the 4 transformed images
> ```
>
> > W2/Q2: Does the PCA method generalize to different vision datasets and tasks? Even different visual (or not) modalities?
>
> Thank you for the question. Yes, PCA can generalize to other vision datasets and tasks. Presented in Table 11, we have already validated PCA on Object Detection using the PASCAL VOC dataset. We have provided a concise table of results (**Table B1**) at your convience:
>
> **Table B1: Results on Object Detection (PASCAL VOC 2007 + 2012)**
> | Method | #. Images | mAP | AP₇₅ | AP₅₀ |
> |---|---|---|---|---|
> | Random | 1,000 | 21.17 ± 0.61 | 17.26 ± 0.85 | 44.97 ± 0.78 |
> | AUM* | 1,000 | 22.31 ± 0.04 | 18.23 ± 0.53 | 46.36 ± 0.39 |
> | VPS[a] | 1,655 | 30.05 ± NaN | 25.84 ± NaN | 60.77 ± NaN |
> | **PCA (Ours)** | **1,000** | **35.99 ± 0.33** | **34.86 ± 0.59** | **65.89 ± 0.44** |
> | Full Dataset | 16,551 | 52.16 | 56.92 | 80.63 |
>
>
> PCA significantly outperformed the current SOTA pruning method (VPS [a]) for detection while using fewer images.
>
> [a] Extending Dataset Pruning to Object Detection: A Variance-based Approach.
>
> > W3/Q3: What happens with transformer backbones? It is noted in the paper that transformer backbones are extremely data-hungry; does this mean that you predict in general that this PCA method will not be effective for transformers? Will it at least fare better than competitors?
>
> **PCA remain effective for transformer backbones**:
>
> 1. **Even though transformers are data-hungry, PCA remains effective.** Transformer-based architectures lack local inductive biases, they underperform CNNs at low IPC but progressively close the gap as dataset size grows. As shown in Table B2, Swin-V2-Tiny follows exactly this trend on PCA-distilled data, surpassing MobileNetV2 and ResNet18 at IPC100. Further improvements can be observed with Hybrid CNN+Transfromer networks (i.e., MaxViT [b]).
>
> **Table B2 (Short version of Table 6): Cross-Architecture Performance.**
> |Type| Backbone | IPC10 | IPC50 | IPC100 |
> |---|---|---|---|---|
> | CNN | ResNet-18 | 22.8 | 39.1 | 45.5 |
> | CNN | MobileNet-V2 | 21.9 | 39.1 | 45.3 |
> | CNN | EfficientNet-B0 | 25.0 | 42.4 | 50.4 |
> | Transformer | Swin-V2-Tiny | 15.3 | 37.8 | 48.2 |
> | Transformer | MaxViT-Tiny [b] | 16.8 | 40.6 | 51.0 |
>
> 2. **On transformer backbones, PCA outperforms other DD/DP methods.** As shown in the table below, at IPC10, competing methods nearly fail entirely on Swin-V2-Tiny, while PCA retains strong performance:
>
> **Table B3: Performance against other SOTA methods on Swin-V2-Tiny, IPC10.**
> | Method | G-VBSM | DWA | EL2N | AUM | PCA |
> | - | - | - | - | - | - |
> | Type | Distillation | Distillation | Pruing | Pruning | Ours |
> | Acc. | 0.3 | 1.4 | 7.7 | 8.6 | **15.3** |
>
> [b] MaxViT: Multi-Axis Vision Transformer.

---

> > ### Author Rebuttal · Reviewer_eDsu · 2026-04-03
> >
> > Thanks for the clarifications, they answered all questions I have. I continue to recommend acceptance.

---

> > > ### Author Response · Authors · 2026-04-03
> > >
> > > Dear Reviewer eDsu,
> > >
> > > We sincerely appreciate your positive assessment. Your constructive feedback during this discussion has been instrumental in clarifying the core contributions of our work.

---

### Official Review · Reviewer_jfCU · 2026-03-13

**Soundness:** 3
**Presentation:** 2
**Significance:** 2
**Originality:** 2
**Overall Recommendation:** 4
**Confidence:** 3

**Summary:**

- This paper aims to fix the label storage issue common with most DD methods. Specifically, labels often require much more storage than the images themselves
- The paper studies the performance of different DD algorithms in the hard label setting, and show that many DD methods have significantly worse performance with hard labels, whereas pruning methods are more robust
- This leads to the PCA algorithm, which essentially combines the cropping combination trick from RDED with a standard pruning algorithm, and with a modified data augmentation scheme that doesn't mix images that are stitched together during training
- Performance is strong (in the hard label setting) compared to baselines

**Compliance With Llm Reviewing Policy:**

Affirmed.

**Final Justification:**

The rebuttal addressed my concerns.

**Key Questions For Authors:**

1. The authors highlight the need for balanced pruning. In table 2 are the pruning algorithms not run with balanced pruning? (if they are not, perhaps the IPC-N description is confusing)
2. Given the weakness point 1, can performance be further improved by stitching 9 downsampled pruned images together?
3. Figure 3 probably should be a bar plot. The dots between data distillation methods on the plot shouldn't be connected to one another.

**Limitations:**

See weaknesses/questions

**Strengths And Weaknesses:**

Strengths
- The issue of DD label storage size is particularly pressing, as with the cost of label storage, DD datasets can be as large as the original dataset, drastically reducing their value. I think tackling this issue is a legitimate challenge and appreciate the work in this area
- The paper very effectively shows the issue with existing DD algorithms and their reliance on soft labels and the contrast with pruning methods in this area is convincing
- Performance is quite strong for a relatively simple algorithm, and there are very large number of experiments
- I appreciate the appendix D.9. detailed comparison to RDED, as at first glance this method seems to have limited novelty compared to it, but the appendix section resolves some of these concerns.

Weaknesses
- Algorithmic/technical novelty is rather low. Especially considering the modified data augmentation scheme to prevent mixed image crops, the algorithm essentially functions as a regular pruning algorithm with 4x the number of images (albeit downsampled). I would appreciate if the authors could clarify this or if I am misunderstanding something
- Presentation is rather cluttered, particularly in the results section. For example pages 7 and 8 just seem like a ton of tables thrown at a page. I think fewer tables in the main text would be better, to highlight important contributions, and the remainder could be in the appendix. Right now with so many tables, it is difficult to parse which ones are central to the paper and which ones are just additional auxiliary results.

---

> ### Author Rebuttal · Authors · 2026-03-30
>
> Thank you for your time and the detailed review. We appreciate the constructive feedback and would like to address your concerns one by one.
>
> > **Q1: Clarify on the Algorithmic/technical novelty.**
>
> Thank you for the question. We respectfully clarify that PCA's novelty is not in any single component, but in the **paradigm shift** from soft-label exploitation to hard-label image-quality-focused compression.
> We highlight three aspects:
>
> **1. The benchmark findings (Sec 3) are a co-equal contribution.**
> We reveal that SOTA DD methods fail to outperform random baselines under soft labels (Tab 2a), fundamentally questioning previous large-scale DD research. This motivates the entire framework.
>
> **2. Each component has theoretical or empirical grounding, not just heuristic design.**
> - **Cropping-free combination** is justified by Prop 4.1 (NLL ↓ ⇏ entropy ↓) and Thm 4.2 (entropy advantage lost under augmentation). This is not a trivial collage: Tab 8 confirms that cropping a well-pruned dataset always hurts.
> - **Constrained augmentation** is the largest single contributor (+8.4% at IPC10, Tab 5). It prevents pruned "easy" images from becoming "hard" during training, aligning with data-scaling laws.
>
> **3. PCA is not simply "pruning with 4× images."**
>
> During training, only **one** sub-image is used per epoch via constrained augmentation, so there is no 4× data advantage. Furthermore, the combination factor ablation (Tab A4, see Q4) shows that larger N does not always help: factor=4 consistently underperforms, confirming that constrained augmentation is a non-trivial design choice.
>
> In addition, we have summarized our contributions in the following tables:
>
> **Table A1: Pruning (Sec 4.1)** *E=Empirical, T=Theoretical*
>
> | Question | Findings | Type | Ref |
> |----------|---------|------|-----|
> | Why use? | Theoretical advantage | T | Fig 4, Lem A.4 |
> | | Performance | E | Fig 3, Tab 2 |
> | Why it works? | Implicit stratification | E | App D.6 |
> | Which method? | EL2N best trade-off | E | Tab 3, App D.2 |
> | | Forgetting limits | E | App D.2 |
> | | Other compatibility | E | Tab 10 |
>
> **Table A2: Combine (Sec 4.2)** *E=Empirical, T=Theoretical*
>
> | Question | Findings | Type | Ref |
> |----------|---------|------|-----|
> | Why cropping-free? | NLL vs. entropy decoupling | T | Prop 4.1 |
> | | Compounding loss under augmentation | T | Thm 4.2 |
> | | Dataset crop impact | E | Tab 8 |
> | | Combination factor ablation | E | Tab A4 (See Q4) |
>
> **Table A3: Augmentation (Sec 4.3)** *E=Empirical, T=Theoretical*
>
> | Question | Findings | Type | Ref |
> |----------|---------|------|-----|
> | Why constrained? | Compounding loss under training crop | T | Thm 4.2 |
> | | Data-scaling law | E | Tab 5 |
> | | Training crop ratio impact | E | Tab 9 |
> | Other options? | Regularization-based | E | Tab 15 |
>
> > Q2: Presentation is rather cluttered, particularly in the results section.
>
> We appreciate the constructive feedback. In the final version, we will move auxiliary results (for example Table 4 and 7) to the Appendix. This will allow us to prioritize a more important contribution in the main text.
>
> > Q3: Are the pruning algorithms in Table 2 run with balanced pruning?
>
> - Yes, we use balanced pruning in Table 2. Actually, we include a comparison of different pruning settings in Table 3, and we use the best setting (balanced and easy) as baselines.
> - We observed that without this constraint, pruning metrics often aggressively remove entire classes at extreme ratios, which would naturally degrade performance.
>
> > Q4: Can performance be further improved by stitching 9 downsampled pruned images together (3x3)?
>
> This is an interesting question. We found the performance improvenment is **not stable** as shown in below table. We ran this experiment across four architectures at IPC=10, varying combination factor $N \in \{2, 3, 4\}$ (i.e., 4, 9, 16 sub-images). We choose N=2 for a balance of image diversity and image quality.
>
> **Table A4: Results of Different Combine Factors.**
> | Factor | ResNet-18 | EfficientNet-B0 | MobileNet-V2 | Swin-Tiny-V2 | Avg |
> |--------|-----------|-----------------|--------------|--------------|-----|
> | 2 (2×2) | 22.8 | 25.9 | 21.9 | **15.3** | **21.3** |
> | 3 (3×3) | **24.6** | **26.2** | **23.3** | 10.9 | 21.2 |
> | 4 (4×4) | 22.3 | 25.7 | 21.9 | 10.0 | 20.0 |
>
>
> > Q5: Figure 3 should be a bar plot rather than a line plot connecting independent DD methods.
>
> Thank you for this suggestion. We agree that a bar plot more accurately and we will update Figure 3 in the revision. We appreciate the reviewer's attention to visualization accuracy, which improves the clarity of our paper.

---

> > ### Author Rebuttal · Reviewer_jfCU · 2026-04-04
> >
> > Thanks for pointing out the difference between PCA vs. more lower resolution images.

---

> > > ### Author Response · Authors · 2026-04-04
> > >
> > > Dear Reviewer jfCU,
> > >
> > > Thank you for your constructive feedback during the discussion. We are pleased that our responses successfully addressed your concerns.

---

### Decision · Program_Chairs · 2026-04-30

**Decision:**

Accept (regular)

**Comment:**

This paper considers the relationship between dataset pruning (DP -- subsample datapoints) and dataset distillation (DD -- convert into a much smaller synthetic dataset) in the context of image classification tasks.

They propose a benchmark for these methods, by mixing real and synthetic images in different proportions on a sliding scale, and testing the use of hard vs soft labels using standard protocols. An important finding is that DD's empirical advantage is driven by soft labels (probability distributions rather than hard labels), which poses serious storage issues as these labels may require more storage than images themselves.

This motivates the authors to introduce a new framework for dataset compression using *hard labels* they call PCA (Prune, Combine, Augment) and demonstrate strong performance in this setting.

All reviewers find the problem well motivated, and agree that the paper provides useful empirical evidence regarding the role of soft labels in dataset distillation. The proposed method is simple and effective.

The main concern was related to the novelty, since the reviewers pointed out that individual components of PCA are relatively standard. Reviewers also inquired why experiments on smaller datasets were not provided, such as CIFAR-10/CIFAR-100/Tiny-ImageNet. The authors successfully clarified that in these cases soft label storage is not an issue, but promissed further experiments in the revision.

Overall, this is a solid and well motivated paper, which provides a timely contribution. While the technical novelty is incremental, the findings and the proposed benchmark are quite relevant, and are likely to be useful to the dataset compression community.